# Tissue-specific modifier alleles determine *Mertk* loss-of-function traits

Yemsratch T Akalu[1†], Maria E Mercau[1†], Marleen Ansems[1], Lindsey D Hughes[1], James Nevin[1], Emily J Alberto[1], Xinran N Liu[2], Li-Zhen He[3], Diego Alvarado[3], Tibor Keler[3], Yong Kong[4], William M Philbrick[5], Marcus Bosenberg[6], Silvia C Finnemann[7], Antonio Iavarone[8], Anna Lasorella[9], Carla V Rothlin[10]*, Sourav Ghosh[11]*

[1]Department of Immunobiology, Yale School of Medicine, New Haven, United States; [2]Department of Cell Biology, Center for Cellular and Molecular Imaging, Yale School of Medicine, New Haven, United States; [3]Celldex Therapeutics, New Haven, United States; [4]Department of Molecular Biophysics and Biochemistry, W. M. Keck Foundation Biotechnology Resource Laboratory, School of Medicine, Yale University, New Haven, United States; [5]Center on Endocrinology and Metabolism, Yale Genome Editing Center, School of Medicine, Yale University, New Haven, United States; [6]Departments of Dermatology, Pathology and Immunobiology, Yale School of Medicine, New Haven, United States; [7]Center for Cancer, Genetic Diseases and Gene Regulation, Department of Biological Sciences, Fordham University, Bronx, United States; [8]Departments of Neurology and Pathology and Cell Biology, Institute for Cancer Genetics, Columbia Medical Center, New York, United States; [9]Departments of Pediatrics and Pathology and Cell Biology, Institute for Cancer Genetics, Columbia University, New York, United States; [10]Departments of Immunobiology and Pharmacology, Yale School of Medicine, New Haven, United States; [11]Departments of Neurology and Pharmacology, Yale School of Medicine, New Haven, United States

*For correspondence:
carla.rothlin@yale.edu (CVR);
sourav.ghosh@yale.edu (SG)

†These authors contributed equally to this work

**Abstract** Knockout (KO) mouse models play critical roles in elucidating biological processes behind disease-associated or disease-resistant traits. As a presumed consequence of gene KO, mice display certain phenotypes. Based on insight into the molecular role of said gene in a biological process, it is inferred that the particular biological process causally underlies the trait. This approach has been crucial towards understanding the basis of pathological and/or advantageous traits associated with *Mertk* KO mice. *Mertk* KO mice suffer from severe, early-onset retinal degeneration. MERTK, expressed in retinal pigment epithelia, is a receptor tyrosine kinase with a critical role in phagocytosis of apoptotic cells or cellular debris. Therefore, early-onset, severe retinal degeneration was described to be a direct consequence of failed MERTK-mediated phagocytosis of photoreceptor outer segments by retinal pigment epithelia. Here, we report that the loss of *Mertk* alone is not sufficient for retinal degeneration. The widely used *Mertk* KO mouse carries multiple coincidental changes in its genome that affect the expression of a number of genes, including the *Mertk* paralog *Tyro3*. Retinal degeneration manifests only when the function of *Tyro3* is concomitantly lost. Furthermore, *Mertk* KO mice display improved anti-tumor immunity. MERTK is expressed in macrophages. Therefore, enhanced anti-tumor immunity was inferred to result from the failure of macrophages to dispose of cancer cell corpses, resulting in a pro-inflammatory tumor microenvironment. The resistance against two syngeneic mouse tumor models observed in *Mertk* KO mice is not, however, phenocopied by the loss of *Mertk* alone. Neither *Tyro3* nor macrophage phagocytosis by alternate genetic redundancy accounts for the absence of anti-tumor immunity. Collectively, our results indicate that context-dependent epistasis of independent modifier alleles determines *Mertk* KO traits.

## Editor's evaluation

In this fundamental contribution, the authors report that a widely used knockout mouse for *Mertk* carries multiple additional changes in its genome, affecting the expression of a number of genes besides *Mertk*. Notably, they show that, although the line was backcrossed to the C57 background, these changes are due to the original 129P2 genome of the embryonic stem cells in which the knockout was originally created and that through the generation of two new knockout mouse strains, in C57 embryonic stem cells, only part of the phenotype of the original *Mertk* knockout mouse can be reproduced. These important and compelling data raise awareness as to the limitations of the *Mertk* [-/- v1] model and limit direct inference of *Mertk* [-/-v1]-observed phenotypes to *Mertk* deficiency alone.

# Introduction

The receptor tyrosine kinase (RTK) MERTK is a paralog of TYRO3 and AXL, and together these receptors are commonly referred to as TAM RTKs. *Mertk* was named after its expression pattern in *m*onocytes, *e*pithelial tissues and *r*eproductive tissues and for it being a *t*yrosine *k*inase (***Graham et al., 1994***). An understanding of MERTK's role in molecular and cellular processes, as well as its broader role in mammalian physiology and pathology, in large part, came from the generation of a *Mertk* knockout (*Mertk* [-/-]) mouse line established by ***Camenisch et al., 1999***. Use of this *Mertk* [-/-] mouse line revealed the critical functional role of this RTK in downregulation of inflammatory cytokines such as TNFα, as well as in the phagocytosis and clearance of apoptotic thymocytes (***Camenisch et al., 1999***; ***Scott et al., 2001***). Subsequently, the *Mertk* [-/-] mouse line became the fountainhead for the description of *Mertk* function in a spectrum of phenotypes spanning retinal degeneration, defective adult neurogenesis, neurodegenerative diseases, liver injury, lupus-like autoimmunity, and cancer (***Cohen et al., 2002***; ***Cook et al., 2013***; ***Crittenden et al., 2016***; ***Davra et al., 2021***; ***Duncan et al., 2003a***; ***Fourgeaud et al., 2016***; ***Huang et al., 2021***; ***Ji et al., 2013***; ***Lindsay et al., 2021***; ***Stanford et al., 2014***; ***Tormoen et al., 2020***; ***Zagórska et al., 2020***).

The *Mertk* [-/-] mouse line was generated by using the available technology of the time. Specifically, *Mertk* was targeted in 129P2/OlaHsd (129P2)-derived E14TG2a embryonic stem (ES) cells (***Camenisch et al., 1999***). ES cells were then microinjected into a C57BL/6 (B6) blastocyst to generate a chimeric mouse with germline transmission of the targeted allele (***Figure 1A and B***). Subsequently, the chimeric mouse was backcrossed to B6 to obtain *Mertk* [-/-] mice, henceforth referred to as *Mertk* [-/- V1] (***Figure 1A***). The *Mertk* [-/-V1] mouse line is available through The Jackson Laboratory (strain# 011122). It is typically backcrossed >10 generations into B6 mice by researchers, including us, and has remained the mainstay for MERTK research. Nevertheless, there have been occasional and isolated reports of independently generated *Mertk* knockout mice that failed to completely recapitulate *Mertk* [-/-V1] phenotypes (***Maddox et al., 2011***). For example, early-onset, severe photoreceptor (PR) degeneration was reported in *Mertk* [-/-V1] mice (***Duncan et al., 2003a***; ***Duncan et al., 2003b***; ***Prasad et al., 2006***). In these mice, the outer nuclear layer (ONL) thickness was significantly reduced by postnatal day (P) 25 (***Duncan et al., 2003a***). Electroretinogram (ERG) recordings revealed that scotopic a- and b-wave amplitudes were significantly lower in *Mertk* [-/- V1] mice at P20 compared to wildtype (WT) mice at P30 (***Duncan et al., 2003a***). Photopic amplitudes were also significantly lower in *Mertk* [-/-V1] mice versus WT mice at P33 (***Duncan et al., 2003a***). An independently generated ENU-induced *Mertk* mutation (*Mertk* [nmf12 or H716R]) in B6 mice caused the substitution of a highly conserved histidine to an arginine and led to a drastic reduction of MERTK in mouse retinas (***Maddox et al., 2011***). Yet it did not identically phenocopy the *Mertk* [-/-V1]-associated early-onset, severe retinal degeneration. Since a slow form of retinal degeneration did indeed occur in *Mertk* [nmf12 or H716R] and MERTK expression was not entirely abolished (***Maddox et al., 2011***), potential problems with *Mertk* [-/-V1] mice were not immediately brought to the fore.

In another independent study, Vollrath et al. demonstrated that crossing *Mertk* [-/-V1] mice to B6 mice occasionally gave rise to animals with normal retina (***Vollrath et al., 2015***). The authors further demonstrated that the *Mertk* [-/-V1] mice carry an ~40 cM chromosomal segment around the *Mertk* locus derived from the 129P2 mouse strain. The very low frequency of normal retina phenotype indicates that crossovers are extremely rare within this chromosomal segment around *Mertk*. Nonetheless, after rare crossover of B6 alleles within this region, *Mertk* [-/-V1]-dependent retinal degeneration was

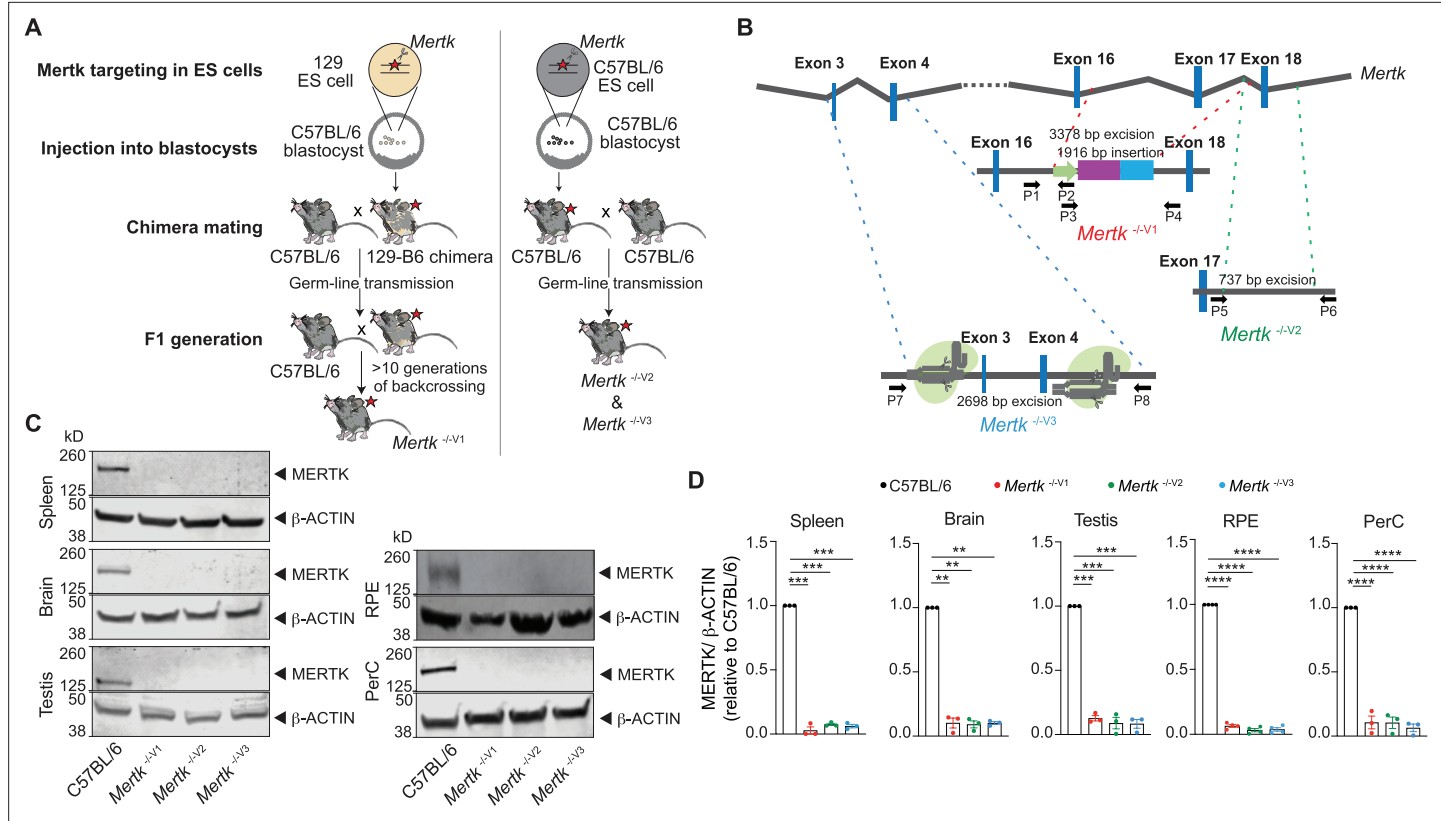

**Figure 1.** Generation of B6 embryonic stem (ES) cell-derived mice with genetic ablation of *Mertk*. (**A**) Schematic showing the differences in approach between the generation of *Mertk* ⁻/⁻ᵛ¹ mice by targeting *Mertk* in 129P2/OlaHsd (129P2)-derived ES cells by Camenisch et al. and our *Mertk* knockout mouse lines. 129P2 ES cells were microinjected into C57BL/6 (B6) blastocysts to generate chimeric mice with germline transmission of deleted *Mertk* allele by Camenisch et al. Chimeric mice were subsequently backcrossed onto B6 mice for >10 generations in our laboratory to obtain *Mertk* ⁻/⁻ᵛ¹ mice. Our two independent *Mertk* knockout mouse lines, *Mertk* ⁻/⁻ᵛ² and *Mertk* ⁻/⁻ᵛ³ mice, were generated by targeting *Mertk* in B6 ES cells. Red stars indicate at least one copy of the mutant allele of *Mertk*. (**B**) Schematic indicating *Mertk* ⁻/⁻ᵛ¹ mice have deletion of exon 17 that encodes for the kinase domain of *Mertk*. A neomycin cassette is also present at this site. *Mertk* ⁻/⁻ᵛ² mice have targeted excision of exon 18, which also encodes for residues in the kinase domain. *Mertk* ⁻/⁻ᵛ³ mice have exons 3 and 4 targeted with CRISPR/Cas9 approach. (**C, D**) Representative and quantification of independent MERTK Western blot data depicting total MERTK protein expression in spleen, brain, testis, retinal pigment epithelia (RPE), and peritoneal cavity cells (PerC) from C57BL/6, *Mertk* ⁻/⁻ᵛ¹, *Mertk* ⁻/⁻ᵛ², and *Mertk* ⁻/⁻ᵛ³ mice (mean ± SEM of n = 3–4 mice/genotype). **p<0.01. ***p<0.001, ****p<0.0001, one-way ANOVA Dunnett's test. Source files for the representative Western blot images (**C**) and the corresponding quantitative analysis (**D**) performed are available in *Figure 1—source data 1*. Supporting data for (**B**) is available in *Figure 1—figure supplement 1A–C*.

The online version of this article includes the following source data and figure supplement(s) for figure 1:

**Source data 1.** Independent datasets and unmodified images for blots shown in *Figure 1*.

**Figure supplement 1.** PCR genotyping of mouse lines targeting *Mertk* (including a mouse line with the simultaneous ablation of *Mertk* and *Tyro3*).

**Figure supplement 1—source data 1.** Unmodified images for PCR genotyping results shown in *Figure 1—figure supplement 1*.

---

prevented (*Vollrath et al., 2015*). Vollrath et al. mapped the suppressor of retinal degeneration to a region that encoded 53 known or predicted open-reading frames (ORFs). *Tyro3* was identified a priori as the likely candidate providing the suppressor function since it is a paralog of *Mertk*. Consistent with this hypothesis, it was observed that TYRO3 expression was at ~33% for *Tyro3* ¹²⁹/¹²⁹ (e.g., in *Mertk* ⁻/⁻ ᵛ¹) relative to *Tyro3* ᴮ⁶/ᴮ⁶ amounts, and associated with retinal degeneration (*Vollrath et al., 2015*). By contrast, TYRO3 expression was at ~67% for *Tyro3* ᴮ⁶/¹²⁹ relative to *Tyro3* ᴮ⁶/ᴮ⁶ amounts. This ~67% expression of TYRO3 prevented retinal degeneration.

These results indicate that not all phenotypes in *Mertk* ⁻/⁻ᵛ¹ mice are solely due to the loss of MERTK function. Such crucial anomalies notwithstanding, the *Mertk* ⁻/⁻ᵛ¹ mouse line continues to be used to ascribe pivotal functions to MERTK in wide-ranging diseases such as neurodegeneration and cancer. We investigated whether indeed phenotypes observed in *Mertk* ⁻/⁻ᵛ¹ mice can be solely and unambiguously ascribed to the loss of function of *Mertk*. Here, we show that two independently generated

B6 *Mertk*[-/-] mouse lines do not phenocopy the retinal degeneration characteristic of *Mertk* [-/-V1] mice. Furthermore, complementary to the genetic evidence that retinal degeneration segregated with *Tyro3* [129/129] but not *Tyro3* [129/B6] reported by Vollrath et al., we demonstrate that the simultaneous ablation of *Mertk* and *Tyro3* in B6 mice is necessary and sufficient for retinal degeneration. MERTK and TYRO3 share the two most well-described TAM functions – phagocytosis and anti-inflammatory signaling (*Rothlin et al., 2015*). Thus, functional redundancy provided by TYRO3 for the phagocytosis of photoreceptor outer segments (POS) by retinal pigment epithelia (RPE) is still consistent with the well-understood mechanism of retinal homeostasis. Interestingly, the B6 *Mertk* knockout mouse lines also did not phenocopy the anti-tumor resistance displayed by *Mertk* [-/-V1] mice against two syngeneic cancer lines. Even mice with simultaneous ablation of *Mertk* and *Tyro3* did not phenocopy the anti-tumor resistance of *Mertk* [-/-V1] mice. Loss of MERTK is proposed to hinder macrophage-dependent phagocytosis of dead or dying cancer cells (efferocytosis). This deficiency in macrophage-mediated disposal of tumor cells, in turn, is postulated to improve availability of tumor antigens for proficient presentation on dendritic cells (DCs) and/or render the tumor microenvironment pro-inflammatory and less immunosuppressive (*Davra et al., 2021*; *Lin et al., 2022*; *Stanford et al., 2014*). Paradoxical to this view, macrophages from *Mertk* [-/-V1], *Mertk* [-/-V2], or *Mertk* [-/-V3] mice all displayed a significant deficit in efferocytosis. RNA sequencing of RPE and bone marrow-derived macrophages (BMDMs) revealed changes in expression of ~12–16 genes located in chromosome 2 in *Mertk* [-/- V1] but not *Mertk* [-/- V2] or *Mertk* [-/- V3] mice. Changes in the expression of additional nonlinked genes beyond chromosome 2 were also observed in *Mertk* [-/-V1] but not *Mertk* [-/-V2] or *Mertk* [-/-V3] mice. This differential gene expression between *Mertk* [-/-V1] and *Mertk* [-/-V2] or *Mertk* [-/-V3] mice was tissue-specific, pointing to the presence of a number of modifier alleles that may function combinatorially in several cell types, and in a variety of biological processes including phagocytosis and beyond, for at least some of the *Mertk* [-/- V1] mouse traits.

## Results

### *Mertk* ablation in B6 ES cells is not sufficient to cause retinal degeneration

We engineered two new *Mertk* knockout mouse lines (designated *Mertk* [-/-V2] and *Mertk* [-/-V3] mice) generated directly using B6 ES cells (*Figure 1A and B*). In the first strategy, we ablated exon 18 within the region encoding the kinase domain and containing the critical ATP-coordinating lysine residue (*Mertk* [-/-V2] mice; *Figure 1B*, *Figure 1—figure supplement 1B*). In an independent approach, we employed CRISPR/CAS9 to delete exons 3 and 4 of *Mertk* (*Mertk* [-/-V3]; *Figure 1B*, *Figure 1—figure supplement 1C*). Immunoblotting of lysates from a variety of tissues, including the spleen, brain, testis, cells from the RPE, and peritoneal cavity (PerC), validated that no detectable MERTK was observed in *Mertk* [-/-V2] and *Mertk* [-/- V3] mice (*Figure 1C and D*). The reduction in MERTK amounts in *Mertk* [-/-V2] and *Mertk* [-/-V3] mice was comparable to that in *Mertk* [-/-V1] tissues (*Figure 1C and D*).

Next, we investigated whether retinal degeneration characteristic of the *Mertk* [-/-V1] mouse is phenocopied in *Mertk* [-/-V2] and *Mertk* [-/-V3] mice. We performed histological analyses as well as transmission electron microscopy of retinal sections at 6 months of age. As expected, *Mertk* [-/-V1] mice displayed advanced PR loss, evidenced by the presence of approximately one row of nuclei in the ONL across the entire dorsal–ventral axis of the retina (*Figure 2A and B*). By contrast, morphological analysis of retinas from *Mertk* [-/-V2] and *Mertk* [-/-V3] mice demonstrated that the ONL thickness in the medial retina of these mice was not significantly different from that of B6 WT mice (*Figure 2A and B*). Evaluation of the retinal ultrastructure at the interface between RPE and POS confirmed severe PR degeneration in *Mertk* [-/-V1] mice (*Figure 2C*), congruent with previous reports (*Duncan et al., 2003a*). Similar assessment of *Mertk* [-/-V2] and *Mertk* [-/-V3] mice at 6 months of age revealed that they had well-preserved RPE microvilli and POS (*Figure 2C*). Consistent with these histological and ultrastructural findings, retinal function was preserved in 6-month-old *Mertk* [-/-V2] and *Mertk* [-/-V3] mice as assessed by scotopic electroretinogram recordings (ERGs) (*Figure 2D–G*). Light-evoked responses in PRs (a-wave) and inner retinal cells (b-wave) in dark-adapted *Mertk* [-/-V2] and *Mertk* [-/-V3] mice were comparable to those in B6 WT mice when tested at increasing luminance levels (*Figure 2D–G*). *Mertk* [-/-V1] mice displayed barely any retinal response to light, which is consistent with earlier studies (*Duncan et al., 2003a*) and the extensive PR

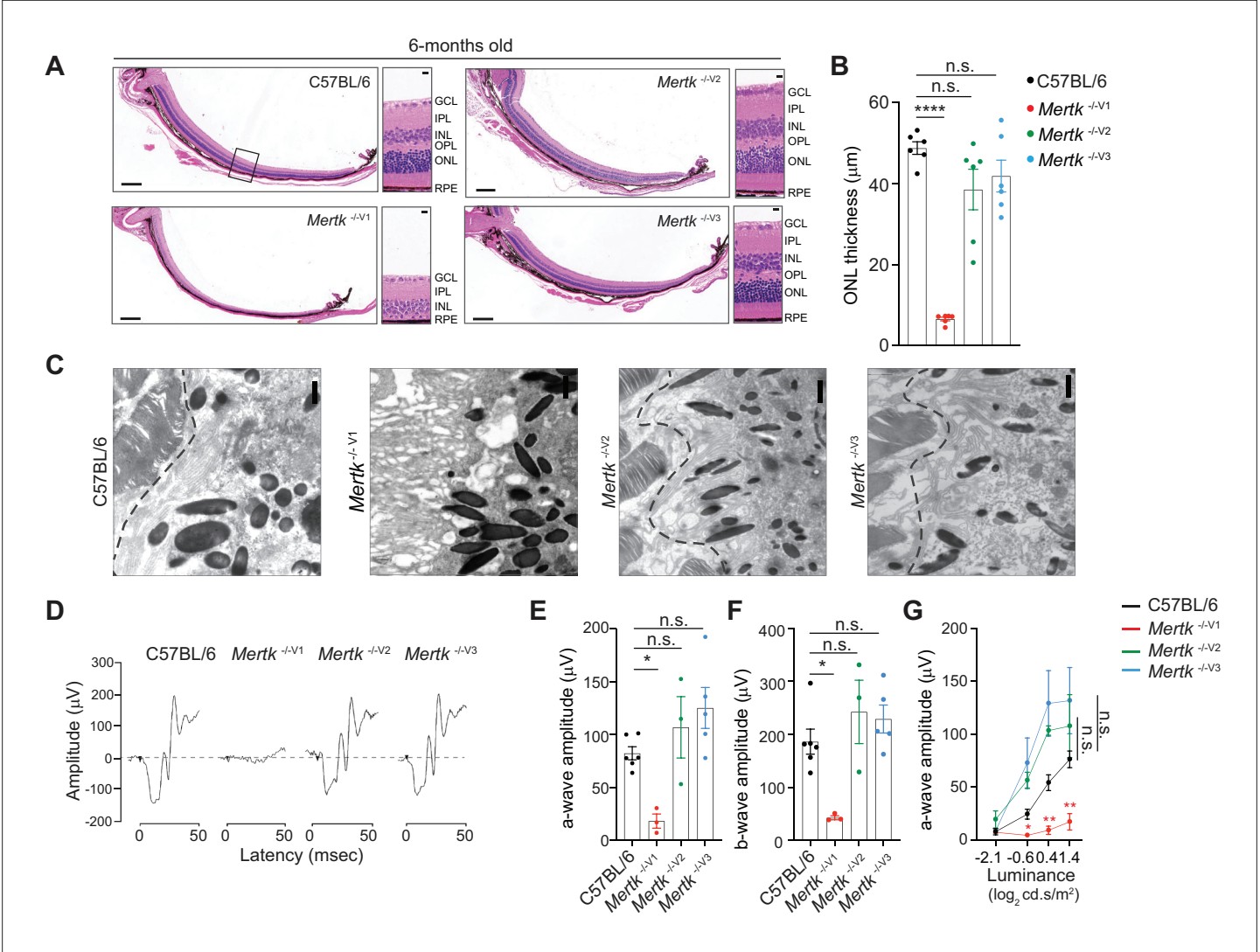

**Figure 2.** Retinal degeneration of *Mertk* [-/-V1] mice is not phenocopied by *Mertk* [-/-V2] and *Mertk* [-/-V3] mice. Morphological and functional changes in the eye were assessed in 6-month-old C57BL/6, *Mertk* [-/-V1], *Mertk* [-/- V2], and *Mertk* [-/-V3] mice. (**A**) Representative hematoxylin-eosin-stained transverse sections of the retina. Boxed section is shown as inset and indicates the area quantified in (**B**). Scale bars = 200 mm (left panels) and 10 mm (insets). (**B**) Quantification of outer nuclear layer (ONL) thickness in the area indicated in (**A**) (mean ± SEM of 10 measurements/mouse, n = 6 mice/genotype). ****p<0.0001, one-way ANOVA Dunnett's test. (**C**) Ultrastructure at the photoreceptor outer segment–retinal pigment epithelia (POS–RPE) interface (dashed line) by transmission electron microscopy. Scale bars = 1 mm. (**D**) Representative scotopic electroretinogram traces are shown at the highest luminance tested. (**E**) Quantification of a-wave amplitude at highest luminance tested (25 cd.s/m²) (mean ± SEM of n = 3–6 mice/ genotype). *p<0.05, one-way ANOVA Dunnett's test. (**F**) Quantification of b-wave amplitude at the highest luminance tested (25 cd.s/m²) (mean ± SEM of n = 3–6 mice/ genotype). *p<0.05, one-way ANOVA Dunnett's test. (**G**) a-wave amplitude at increasing luminance (mean ± SEM of n = 3–6 mice/ genotype). *p<0.05, **p<0.01, two-way ANOVA. Source files for (**B**) ONL thickness, (**E**) a-wave amplitude, (**F**) b-wave amplitude, and (**G**) a-wave amplitude at increasing luminance are available in *Figure 2—source data 1*. GCL,-ganglion cell layer; IPL, inner plexiform layer; INL, inner nuclear layer; OPL, outer plexiform layer.

The online version of this article includes the following source data for figure 2:

**Source data 1.** Quantification of ONL thickness, a-wave and b-wave amplitude for data shown in *Figure 2*.

degeneration observed (*Figure 2D–G*). Thus, the retinal degeneration characteristic of *Mertk* $^{-/-V1}$ mice is not phenocopied by knocking out *Mertk* in B6 ES cell-derived mice (*Mertk* $^{-/-V2}$ and *Mertk* $^{-/-V3}$ mice).

## Gene expression differences in *Mertk* $^{-/-}$ RPE revealed by genome-wide transcriptional analyses in the presence of 129P2 versus B6 alleles on chromosome 2

A segment of chromosome 2 in the *Mertk* $^{-/-V1}$ mice was previously reported to be derived from the 129P2 background (*Vollrath et al., 2015*). Therefore, we performed short tandem repeat (STR) analysis in the chromosome 2 region surrounding the *Mertk* locus in B6 WT, *Mertk* $^{-/-V1}$, *Mertk* $^{-/-V2}$, and *Mertk* $^{-/-V3}$ mice. Genomic DNA isolated from each of these mouse lines was subjected to PCR

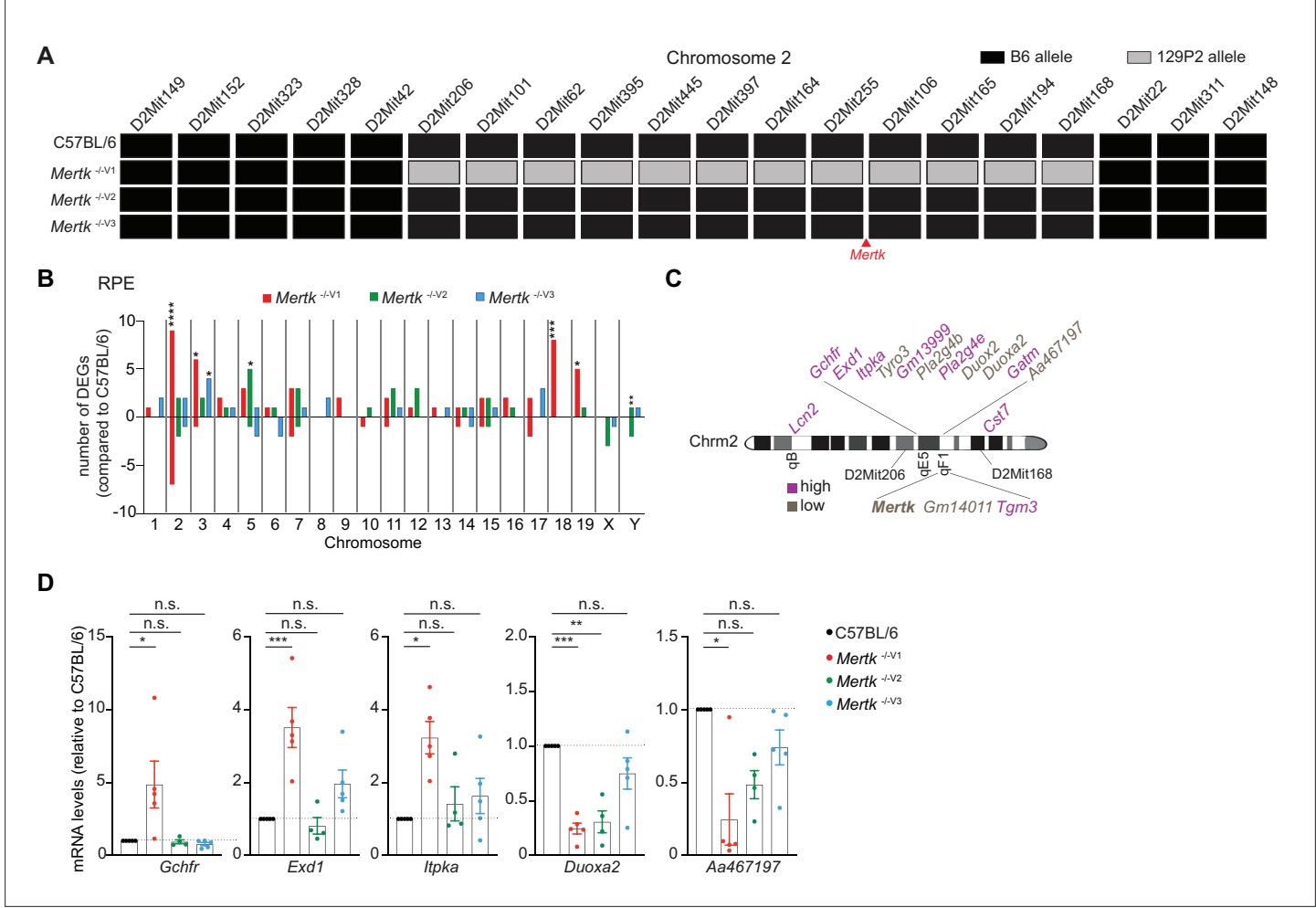

**Figure 3.** Significant gene expression differences in retinal pigment epithelia (RPE) of *Mertk* $^{-/-V1}$ versus *Mertk* $^{-/-V2}$ and *Mertk* $^{-/-V3}$ mice. (**A**) Haplotype map comparing 20 microsatellite markers across chromosome 2 in C57BL/6, *Mertk* $^{-/-V1}$, *Mertk* $^{-/-V2}$, and *Mertk* $^{-/-V3}$ mice. Black rectangles indicate homozygosity for C57BL/6 (B6) alleles, and gray rectangles indicate homozygosity for 129P2/OlaHsd (129P2) alleles. (**B**) Distribution of differentially expressed genes (DEGs) in RPE across chromosomes (n = 3–6 samples/genotype). *p<0.05, **p<0.01, ***p<0.001, ****p<0.0001, hypergeometric test. (**C**) Schematic showing genes neighboring *Mertk* that are significantly upregulated or downregulated in the RPE of *Mertk* $^{-/-V1}$ mice. (**D**) qPCR quantification of indicated chromosome 2 genes in C57BL/6, *Mertk* $^{-/-V1}$, *Mertk* $^{-/-V2}$, and *Mertk* $^{-/-V3}$ RPEs (mean ± SEM, n = 4–5 samples/genotype). *p<0.05, **p<0.01, ***p<0.001, one-way ANOVA Dunnett's test. Source files for the distribution of DEGs in RPEs across chromosomes (**B**) and qPCR quantification of various chromosome 2 genes (**D**) are available in *Figure 3—source data 1*. Supporting data for (**A**) is available in *Figure 3—figure supplement 1*.

The online version of this article includes the following source data and figure supplement(s) for figure 3:

**Source data 1.** Independent datasets for qPCR quantifications shown in *Figure 3D*.

**Figure supplement 1.** A 129P2-specific genomic interval is present in *Mertk* $^{-/-V1}$ but not in *Mertk* $^{-/-V2}$ and *Mertk* $^{-/-V3}$ mice.

**Figure supplement 1—source data 1.** Unmodified images for PCR genotyping results shown in *Figure 3—figure supplement 1*.

amplification of 24 microsatellite sites across chromosome 2 (*Figure 3—figure supplement 1*). We found that *Mertk* [-/-V1] mice harbored a 15.08 cM region between D2Mit206 and D2Mit168 that is of 129P2 origin (*Figure 3A*). As expected, chromosome 2 was entirely B6-derived in both *Mertk* [-/-V2] and *Mertk* [-/-V3] mice (*Figure 3A*).

To more broadly understand the genome-wide transcriptional differences between *Mertk* [-/-V1], *Mertk* [-/-V2], and *Mertk* [-/-V3] mice stemming from the chromosomal differences, we performed RNA sequencing (RNAseq) experiments on RPE from these three mouse lines and control B6 WT RPE at P25. We identified a number of genes that were differentially expressed in *Mertk* [-/-V1], *Mertk* [-/-V2], and *Mertk* [-/-V3] RPE in *cis* and in *trans* compared to B6 WT RPE (*Figure 3B and C*). Significant changes in the transcripts of *Mertk*-neighboring genes, in *Mertk* [-/-V1], but not *Mertk* [-/-V2] and *Mertk* [-/-V3], RPE relative to B6 WT RPE were confirmed by qPCR (*Figure 3D*).

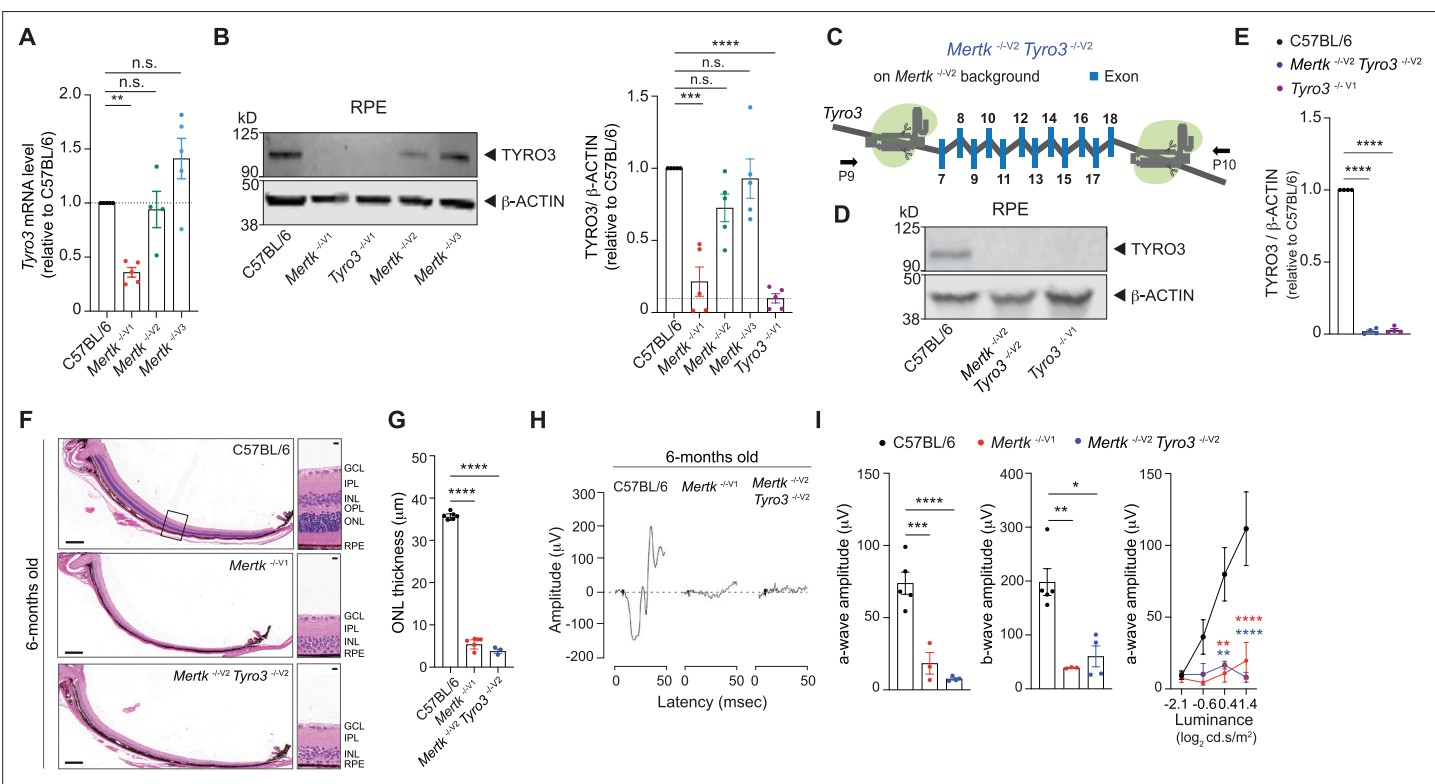

**Figure 4.** *Tyro3* is epistatic with *Mertk* for the retinal degeneration trait. (**A**) qPCR quantification of *Tyro3* in C57BL/6, *Mertk* [-/-V1], *Mertk* [-/-V2], and *Mertk* [-/-V3] retinal pigment epithelia (RPE) (mean ± SEM, n = 4–5 samples/genotype). **p<0.01, one-way ANOVA Dunnett's test. (**B**) Representative and independent measurements of TYRO3 amounts in RPE. Western blot (WB) from C57BL/6, *Mertk* [-/-V1], *Mertk* [-/-V2], *Mertk* [-/-V3], and *Tyro3* [-/-V1] mice RPE (mean ± SEM of n = 5 mice/ genotype). *p<0.05, **p<0.01, ***p<0.001, and ****p<0.0001, one-way ANOVA Dunnett's test. (**C**) Schematic showing targeting of *Tyro3* exons 7–18 with CRISPR/Cas9 in *Mertk* [-/-V2] ES cells to generate the *Mertk* [-/-V2] *Tyro3* [-/-V2] mouse line. Image not drawn to scale. (**D, E**) Representative and independent measurements of TYRO3 amounts in RPE. WB from C57BL/6, *Mertk* [-/-V2] *Tyro3* [-/-V2], and *Tyro3* [-/-V1] mice RPE (mean ± SEM of n = 4 mice/ genotype). ****p<0.0001, one-way ANOVA Dunnett's test. (**F**) Representative hematoxylin-eosin-stained transverse sections of the retina. Boxed section is shown as inset and indicates the areas quantified in (**G**). Scale bars = 200 mm (left panels) and 10 mm (insets). (**G**) Quantification of outer nuclear layer (ONL) thickness in the area indicated in (**F**) (mean ± SEM of 10 measurements/mouse, n = 3–6 mice/genotype). ****p<0.0001, one-way ANOVA Dunnett's test. (**H**) Representative scotopic electroretinogram traces are shown at the highest luminance tested. (**I**) Quantification of a-wave amplitude and b-wave amplitude at highest luminance tested (mean ± SEM of n = 3–6 mice/genotype). *p<0.05, **p<0.01, ***p<0.001, ****p<0.0001, one-way ANOVA Dunnett's test. a-wave amplitude at increasing luminances. **p<0.01, ****p<0.0001, two-way ANOVA. Morphological and functional changes in the eye were assessed in 6-month-old C57BL/6, *Mertk* [-/-V1], and *Mertk* [-/-V2] *Tyro3* [-/-V2] mice. Source files for (**A**) qPCR quantification of *Tyro3*, (**B, E**) quantification of TYRO3 levels, (**G**) ONL thickness, (**I**) a-wave amplitude, b-wave amplitude, and a-wave amplitude at increasing luminances are available in *Figure 4—source data 1*. Supporting data for (**C**) is available in *Figure 1—figure supplement 1D*. GCL, ganglion cell layer; IPL, inner plexiform layer; INL, inner nuclear layer; OPL,-outer plexiform layer.

The online version of this article includes the following source data for figure 4:

**Source data 1.** Independent datasets and unmodified images for results shown in *Figure 4*.

### *Tyro3* is epistatic with *Mertk* for the retinal degeneration trait

It was previously reported that TYRO3 levels were significantly lower in *Tyro3* [129/129] RPE compared to *Tyro3* [B6/B6] or *Tyro3* [129/B6] RPE (*Vollrath et al., 2015*). Consistent with this report, we found *Tyro3* mRNA level to be significantly downregulated in *Mertk* [-/-V1], but not in *Mertk* [-/-V2] and *Mertk* [-/-V3], RPE (*Figure 4A*). Moreover, we detected significantly lower levels of TYRO3 in *Mertk* [-/-V1] RPE by Western blot (*Figure 4B*). By contrast, RPE cells from *Mertk* [-/-V2] and *Mertk* [-/-V3] mice had levels of TYRO3 that were comparable to B6 WT mice (*Figure 4B*). Since it was concluded that even the hypomorphic expression of *Tyro3* [B6/129] can suppress the phenotypes of *Mertk* loss of function (*Vollrath et al., 2015*), we tested whether the simultaneous genetic ablation of *Mertk* and *Tyro3* can phenocopy *Mertk* [-/-V1] mice. We engineered *Mertk* [-/-V2] *Tyro3* [-/-V2] mice by targeting *Tyro3* in *Mertk* [-/-V2] mice using CRISPR/CAS9 (*Figure 4C*, *Figure 1—figure supplement 1D*). Immunoblotting experiments confirmed that TYRO3 was undetectable in the RPE of *Mertk* [-/-V2] *Tyro3* [-/-V2] mice when compared to TYRO3 expression in B6 WT RPE (*Figure 4D and E*). Indeed, these mice recapitulated the severe retinal degeneration observed in *Mertk* [-/-V1] underscoring the function of *Tyro3* [B6] as a suppressor allele in retinal degeneration induced by targeting *Mertk*. When eye sections from 6-month-old *Mertk* [-/-V2] *Tyro3* [-/-V2], *Mertk* [-/-V1] and B6 WT controls were stained with hematoxylin-eosin, we found that ONL thickness in *Mertk* [-/-V2] *Tyro3* [-/-V2] mice was significantly reduced compared to B6 WT controls, commensurate with *Mertk* [-/-V1] mice (*Figure 4F and G*). Similar to *Mertk* [-/-V1] mice, *Mertk* [-/-V2] *Tyro3* [-/- V2] mice had only approximately one row of nuclei in the ONL across the entire dorsal–ventral axis of the retina (*Figure 4F and G*). Consequently, *Mertk* [-/-V2] *Tyro3* [-/-V2] mice did not display light-evoked responses in scotopic ERGs at

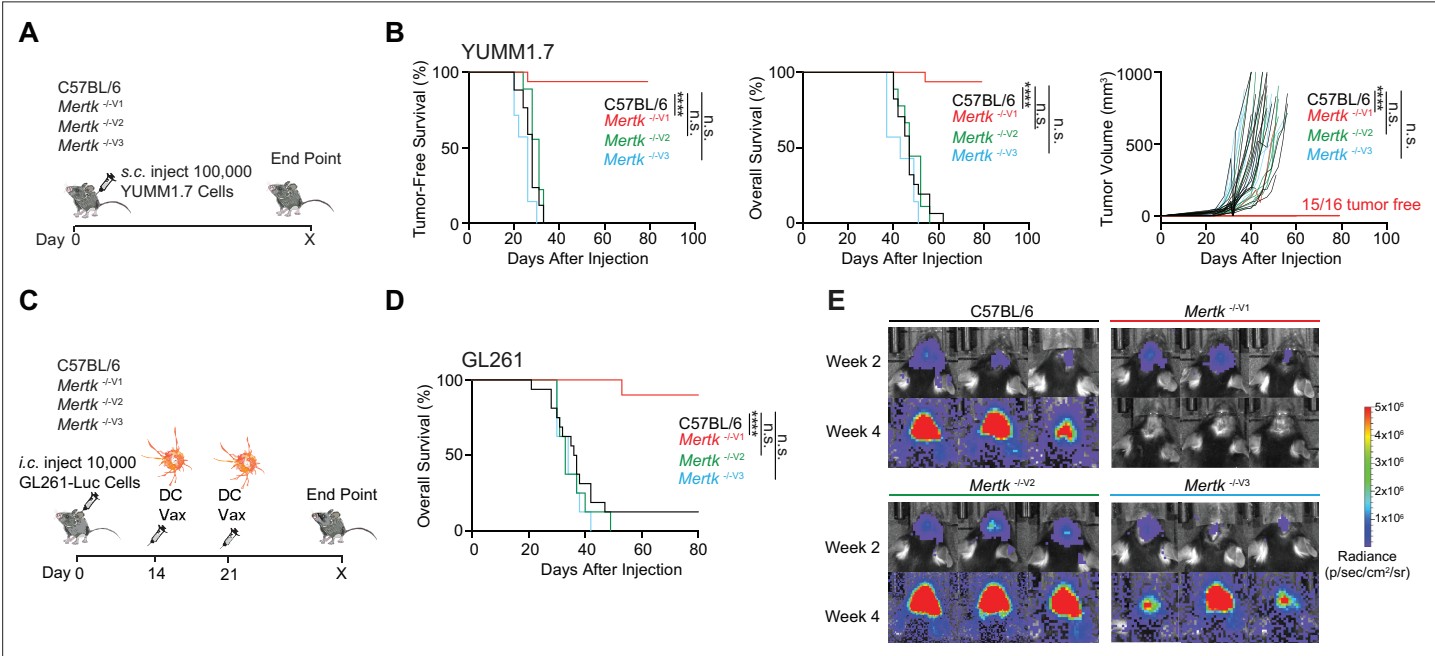

**Figure 5.** Anti-tumor response of *Mertk* [-/-V1] mice is not phenocopied by *Mertk* [-/-V2] and *Mertk* [-/-V3] mice. (**A**) Schematic showing subcutaneous injection of 100,000 *Braf* [V600E] *Pten* [-/-] YUMM1.7 mouse melanoma cells into mice of different genotypes. (**B**) Tumor-free survival (TFS), overall survival (OS), and tumor volume in C57BL/6 (n = 17), *Mertk* [-/-V1] (n = 16), *Mertk* [-/-V2] (n = 9), and *Mertk* [-/-V3] (n = 7) mice implanted with YUMM1.7 cells. ****p<0.0001, Log-rank Mantel–Cox test or two-way ANOVA, Dunnett's multiple-comparison test. (**C**) Schematic showing intracranial injection of 10,000 GL261-Luc glioma cells and intraperitoneal dendritic cell vaccination of mice at 14 and 21 days post-tumor implantation. (**D**) OS in C57BL/6 (n = 16), *Mertk* [-/-V1] (n = 10), *Mertk* [-/-V2] (n = 8), and *Mertk* [-/- V3] (n = 8) mice implanted with GL261 tumors. ****p<0.0001, Log-rank Mantel–Cox test. (**E**) Representative IVIS images of intracranial tumors at D14 and D28 post-implantation. Source files for TFS (**B**) and OS (**B, D**) plots shown are available in *Figure 5—source data 1*. Supporting data relating to (**C–E**) is available in *Figure 5—figure supplement 1*.

The online version of this article includes the following source data and figure supplement(s) for figure 5:

**Source data 1.** Tumor free survival dataset following implantation with YUMM1.7 melanoma cells shown in *Figure 5*.

**Source data 2.** Overall survival dataset following implantation with GL261 glioblastoma cells shown in *Figure 5*.

**Figure supplement 1.** Unvaccinated *Mertk* [-/- V1] mice are not protected against GL261 tumors.

any luminance tested (*Figure 4H and I*). These results show that *Mertk* $^{-/-V2}$ *Tyro3* $^{-/-V2}$ mice phenocopy the widespread morphological and functional retinal deficits characteristic of *Mertk* $^{-/-V1}$ mice.

## *Mertk* targeting in 129P2 ES cells results in a robust anti-tumor response against immune checkpoint inhibitor (ICI)-refractory YUMM1.7 melanoma and GL261 brain tumor, but this effect is not phenocopied in *Mertk* $^{-/-V2}$ or *Mertk* $^{-/-V3}$ mice

An important, newly discovered role of MERTK is as an innate immune checkpoint in cancer (*Cook et al., 2013*; *Davra et al., 2021*; *Huelse et al., 2020*; *Lee-Sherick et al., 2020*; *Lin et al., 2022*; *Lindsay et al., 2021*; *Sekar et al., 2022*; *Su et al., 2020*; *Wu et al., 2018*; *Zhou et al., 2020*). *Mertk* $^{-/-V1}$ mice were used to demonstrate improved anti-tumor immune response against MMTV-PyVmT, B16:F10, and MC38 tumor models (*Cook et al., 2013*). Based on the use of *Mertk* $^{-/-V1}$ mice, it was also surmised that inhibition of macrophage efferocytosis due to MERTK loss of function results in decreased tumor growth and increased tumor-free survival (TFS) in E0771 murine breast cancer model (*Davra et al., 2021*). Similarly, *Mertk* $^{-/-V1}$ mice were used in a study by Lindsay et al. to demonstrate improved anti-tumor T cell motility in a B78ChOva tumor model (*Lindsay et al., 2021*). We compared the anti-tumor response of *Mertk* $^{-/-V1}$, *Mertk* $^{-/-V2}$, and *Mertk* $^{-/-V3}$ mice in a model of ICI-refractory melanoma (YUMM1.7), as well as in an orthotopic brain tumor mouse model (GL261). YUMM1.7 tumor cells were implanted subcutaneously in *Mertk* $^{-/-V1}$, *Mertk* $^{-/-V2}$, and *Mertk* $^{-/-V3}$ mice or in B6 WT (*Figure 5A*). TFS, overall survival (OS), and rate of tumor growth were monitored (*Figure 5A and B*). Consistent with previous reports of MERTK blockade enhancing anti-tumor response (*Cook et al., 2013*; *Davra et al., 2021*; *Huelse et al., 2020*; *Lee-Sherick et al., 2020*; *Lin et al., 2022*; *Lindsay et al., 2021*; *Sekar et al., 2022*; *Su et al., 2020*; *Wu et al., 2018*; *Zhou et al., 2020*), 93.75% of *Mertk* $^{-/-V1}$ mice remained tumor free compared to 0% of B6 WT control mice (*Figure 5B*). Surprisingly, neither of the *Mertk* knockout mice generated using B6 ES cells, *Mertk* $^{-/-V2}$ and *Mertk* $^{-/-V3}$, phenocopied the tumor resistance observed in *Mertk* $^{-/-V1}$ mice. Also, 100% of *Mertk* $^{-/-V2}$ and *Mertk* $^{-/-V3}$ mice succumbed to tumor growth at a rate comparable to their B6 WT counterparts (*Figure 5B*). To investigate these findings in an independent tumor model, luciferase-expressing GL261 brain tumor cells were orthotopically injected in B6 and *Mertk* $^{-/-V1}$ mice. Bioluminescence imaging was performed to determine tumor volume and mice were monitored for OS (*Figure 5—figure supplement 1*). No differences were observed in the rate of GL261 growth or time to end point between B6 WT or *Mertk* $^{-/-V1}$ mice (*Figure 5—figure supplement Figure 5—figure supplement 1B and C*). We next attempted to treat tumor-bearing mice with a dendritic cell vaccine (DC-Vax). Tumor cell lysates obtained by freeze-thawing GL261 were fed to B6 WT bone marrow-derived DCs. Following co-incubation, DCs were activated with LPS treatment and intraperitoneally injected into tumor-bearing B6 WT or *Mertk* $^{-/-V1}$ mice on days 14 and 21 after tumor implantation (*Figure 5C*). Of note, tumor sizes were comparable between B6 WT or *Mertk* $^{-/-V1}$ mice prior to receiving the DC-Vax (*Figure 5E*, top panel). Remarkably, the administration of a DC-Vax in *Mertk* $^{-/-V1}$ mice resulted in a significant reduction in tumor size, whereas DC-Vax-treated B6 WT mice succumbed to their tumor burden (*Figure 5D and E*). Similar to the studies on YUMM1.7 melanoma cells, when GL261 cells were implanted in *Mertk* $^{-/-V2}$ and *Mertk* $^{-/- V3}$ mice, and these mice were subsequently treated with DC-Vax, 100% of these mice failed to display the anti-GL261 response of *Mertk* $^{-/-V1}$ mice (*Figure 5D and E*). Taken together, our experiments revealed an astounding anti-tumor resistance of *Mertk* $^{-/-V1}$ mice against both YUMM1.7 and GL261. However, this phenotype of *Mertk* $^{-/-V1}$ mice was again not phenocopied by *Mertk* $^{-/-V2}$ and *Mertk* $^{-/-V3}$ mice.

## Neither deficient efferocytosis in macrophages nor loss of *Tyro3* $^{B6/B6}$ can universally account for the anti-tumor immunity in *Mertk* $^{-/-V1}$ mice

The prevailing dogma is that deficient phagocytosis of tumor cells by macrophages in the absence of *Mertk* function represents the *sine qua non* of the remarkable anti-tumor immunity. It was proposed that the absence of MERTK in tumor-associated macrophages prevents the phagocytosis and disposal of dead tumor cells, thereby increasing the availability of tumor antigen to DCs and triggering improved anti-tumor T cell immunity (*Davra et al., 2021*). Similarly, it was speculated that deficient MERTK-dependent phagocytosis by tumor-associated macrophages leads to secondary necrosis of tumor cells and an increased pro-inflammatory microenvironment more conducive to anti-tumor

immunity (*Stanford et al., 2014*). It was also suggested that blockade of MERTK-dependent phagocytosis in tumor-associated macrophages activated the STING pathway (*Davra et al., 2021*; *Stanford et al., 2014*; *Zhou et al., 2020*). Of note, these scenarios are not mutually exclusive and might in fact cooperate. We performed similar RNAseq analyses in BMDMs isolated from *Mertk* [-/-V1], *Mertk* [-/-V2] and *Mertk* [-/-V3] and B6 WT mice. Our experiments identified a number of changes in *Mertk* [-/-V1], *Mertk* [-/-V2], and *Mertk* [-/-V3] BMDMs compared to B6 WT controls. Furthermore, *Mertk* [-/-V1] mice had a significant number of upregulated and downregulated genes that were not recapitulated in *Mertk* [-/-V2] and *Mertk* [-/-V3] mice (*Figure 6A–C*). Although the lack of anti-tumor resistance in *Mertk* [-/-V2] and *Mertk* [-/-V3] mice already pointed to a MERTK-agnostic basis for tumor clearance, we hypothesized that some of the coincidental changes in BMDMs in these mice might compensate for the loss of MERTK by providing redundancy in phagocytosis. Thus, we expected no reduction in efferocytosis in BMDMs derived from *Mertk* [-/-V2] and *Mertk* [-/-V3] mice, unlike BMDMs from *Mertk* [-/-V1] mice. We generated CD11b[+] F4/80[+] BMDMs from B6 WT, *Mertk* [-/-V1], *Mertk* [-/-V2], and *Mertk* [-/-V3] mice and tested them in an ex vivo phagocytosis assay. Flow cytometry-based analysis of BMDMs derived from *Mertk* [-/-V2] and *Mertk* [-/-V3] mice confirmed MERTK ablation in these cells compared to B6 controls (*Figure 6—figure supplement Figure 6—figure supplement 1A and B*). BMDMs were cultured with pHrodo-labeled apoptotic thymocytes in the presence of serum and their uptake was assayed by flow cytometry after 1 hr. As expected, *Mertk* [-/-V1] BMDMs were ~50% less phagocytic than B6 WT BMDMs. BMDMs from *Mertk* [-/-V2] and *Mertk* [-/-V3] mice were similarly less phagocytic than those from B6 WT mice (*Figure 6D and E*). These experiments demonstrate that BMDMs derived from *Mertk* knockout mice of 129P2 or B6 origin were equally deficient in the efferocytosis of apoptotic thymocytes.

It is important to note that TYRO3 can not only compensate for MERTK in phagocytosis (*Vollrath et al., 2015*); it is also a negative regulator of the immune response (*Chan et al., 2016*). However, *Tyro3* was not a differentially expressed gene (DEG) in BMDMs because it is not expressed in these cells (*Figure 6A–C*). To entirely rule out *Tyro3* [B6] function as a suppressor in abolishing the anti-tumor effects of *Mertk* ablation in *Mertk* [-/-V2] and *Mertk* [-/-V3] mice, we implanted YUMM1.7 and GL261 tumors in *Mertk* [-/-V2] *Tyro3* [-/-V2] mice. However, we observed that *Mertk* [-/-V2] *Tyro3* [-/-V2] mice entirely failed to phenocopy the anti-tumor resistance observed in *Mertk* [-/-V1] mice (*Figure 6F–H*). Specifically, 100% of *Mertk* [-/-V2] *Tyro3* [-/-V2] mice failed to show improved survival in comparison to B6 WT mice when implanted with YUMM1.7 (*Figure 6F*). No differences were observed in TFS, OS, or in tumor growth (*Figure 6F*). Similarly, in the GL261 model, no differences were observed in OS or in tumor volume in *Mertk* [-/-V2] *Tyro3* [-/-V2] mice following the DC vaccination strategy that conferred significant anti-tumor resistance to *Mertk* [-/-V1] mice (*Figure 6G and H*).

## Tissue-specific discordance in gene expression between *Mertk* [-/-] mouse lines carrying 129P2 versus B6 alleles

How are *Mertk* [-/-V1] mouse traits modified? RNAseq demonstrated that gene expression changes in *Mertk* [-/-V1] mice are not restricted to changes in *cis* on chromosome 2, but expanded in *trans* to other chromosomes. A number of genes displaying differential expression in *Mertk* [-/-V1] versus *Mertk* [-/-V2] and *Mertk* [-/-V3] in BMDMs mapped between D2Mit206 (55.94 cM) and D2Mit168 (71.02 cM) (in *cis*) on chromosome 2 (*Figure 6B*). We were able to confirm that genes mapping to this region, including *Gchfr, Exd1,* and *Gatm,* were increased by ~2.5- to 12-fold in *Mertk* [-/-V1] but not *Mertk* [-/-V2] and *Mertk* [-/-V3] BMDMs, compared to B6 WT BMDMs, by qPCR (*Figure 6C*). Additionally, transcripts such as *Aa467197, Thbs1,* and *Pmepa1* were found to be ~40–80% decreased in *Mertk* [-/- V1] but not *Mertk* [-/-V2] and *Mertk* [-/-V3] BMDMs, compared to B6 WT BMDMs (*Figure 6C*). Interestingly, genes beyond this chromosomal segment (i.e., in *trans*) were also differentially expressed between B6 WT, *Mertk* [-/-V1], *Mertk* [-/- V2], and *Mertk* [-/-V3] BMDMs (*Figure 6A*). The changes in the expression of genes in *trans* common to *Mertk* [-/-V1], *Mertk* [-/-V2], and *Mertk* [-/-V3] BMDMs by comparison to B6 WT are likely direct downstream consequences of *Mertk* ablation. Nevertheless, there were also distinct changes in *trans* in *Mertk* [-/-V1] BMDMs not reproduced in *Mertk* [-/-V2] and *Mertk* [-/-V3] BMDMs (*Figure 6A*). Such changes are likely independent of *Mertk*.

Importantly, many of the DEGs, in *cis* as well as in *trans*, were distinct between BMDMs and RPE in *Mertk* [-/-V1] versus *Mertk* [-/-V2] and *Mertk* [-/-V3] (*Figure 4B–D*, *Figure 6A–C*). For instance, DEGs corresponding to chromosome 2 such as *Itpka, Gm13999, Pla2g4e, Tgm3, Duoxa2,* and *Slc28a2b* were unique to BMDMs or RPE. Similarly, in chromosomes 18 and 19, there were a number of DEGs unique

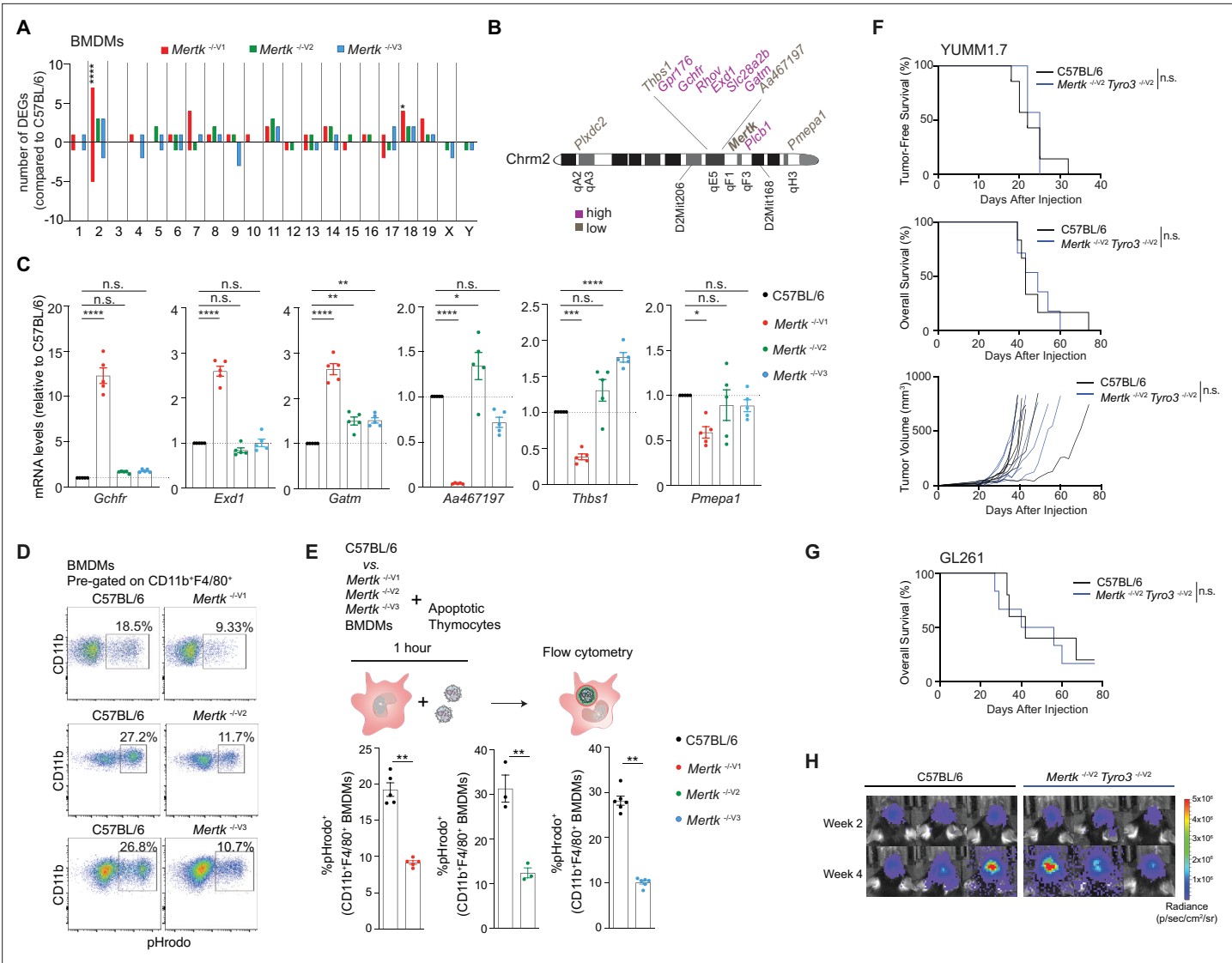

**Figure 6.** Anti-tumor response of *Mertk* [-/-V1] mice is neither the result of deficient efferocytosis by macrophages nor hypomorphic TYRO3. (**A**) Distribution of differentially expressed genes (DEGs) in bone marrow-derived macrophages (BMDMs) across chromosomes. (n = 3 samples/ genotype). *p<0.05, ****p<0.0001, hypergeometric test. (**B**) Schematic indicating chromosome 2 genes that are upregulated or downregulated in *Mertk* [-/-V1] BMDMs. (**C**) qPCR quantification of the indicated chromosome 2 genes in C57BL/6, *Mertk* [-/-V1], *Mertk* [-/-V2], and *Mertk* [-/-V3] BMDMs (mean ± SEM, n = 5 samples/genotype). *p<0.05, **p<0.01, ***p<0.001, ****p<0.0001, one-way ANOVA Dunnett's test. (**D**) BMDMs from C57BL/6, *Mertk* [-/-V1], *Mertk* [-/-V2], and *Mertk* [-/-V3] mice were co-cultured with apoptotic thymocytes for 1 hr. Representative plots show uptake of pHrodo-labeled apoptotic thymocytes by CD11b⁺F4/80⁺ BMDMs by flow cytometry. (**E**) Quantification of (**D**) (mean ± SEM, n = 3–6 mice/group). **p<0.01, Mann–Whitney or Student's *t*-test. (**F**) Tumor-free survival (TFS), overall survival (OS), and tumor volume in C57BL/6 (n = 7), *Mertk* [-/-V1] (n = 7), and *Mertk* [-/-V2] *Tyro3* [-/-V2] (n = 7) mice implanted with YUMM1.7 tumors. n.s., Log-rank Mantel–Cox test. (**G**) OS in C57BL/6 (n = 5) and *Mertk* [-/-V2] *Tyro3* [-/-V2] (n = 6) mice implanted with GL261 tumors. n.s., Log-rank Mantel–Cox test. (**H**) Representative IVIS images of intracranial tumors in C57BL/6 and *Mertk* [-/-V2] *Tyro3* [-/-V2] mice at D14 and D28 post-implantation. Source files for the distribution of DEGs in BMDMs across chromosomes (**A**), qPCR quantification of various chromosome 2 genes (**C**), quantification of percentage of pHrodo⁺ BMDMs (**E**), TFS (**F**), and OS (**F, G**) plots are available in *Figure 6—source data 1*. Supporting data for (**D, E**) is available in *Figure 6—figure supplement 1*.

The online version of this article includes the following source data and figure supplement(s) for figure 6:

**Source data 1.** Independent datasets for qPCR quantifications and survival dataset following implantation with tumor models shown in *Figure 6*.

**Source data 2.** Quantification of MERTK levels in bone-marrow-derived macrophages shown in *Figure 6*.

**Figure supplement 1.** Significantly diminished MERTK expression in bone marrow-derived macrophages (BMDMs) isolated from *Mertk* knockout mice.

to BMDMs or RPE (*Figure 4B*, *Figure 6A*). Therefore, it is likely that there might be tissue-specific modifiers operating both in *cis* and *trans*. Additionally, there exist some genes that are transcribed in both BMDMs and RPE but are differentially expressed in either BMDMs or RPE only. For example, *Lcn2* and *Cst7* were detected as DEGs corresponding to chromosome 2 in *Mertk* [-/-V1] versus *Mertk* [-/-V2] and *Mertk* [-/-V3] RPE, while *Plcb1*, *Gpr176*, *Rhov*, *Thbs1*, and *Plxdc2* were DEGs in BMDMs only. This difference in chromosome 2 DEG identity between *Mertk* [-/-V1] versus *Mertk* [-/-V2] and *Mertk* [-/-V3] across tissues that basally transcribe said genes points to the impact of genetic background-specific regulation of gene expression. Taken together, complex traits in *Mertk* [-/-V1] mice, such as anti-tumor resistance, are likely to result from a combination of cell type- and genetic background-specific changes.

## Discussion

The *Mertk* [-/-V1] mice (*Camenisch et al., 1999*), heretofore, has remained the workhorse for investigation of the biological consequences of *Mertk* ablation. Based on studies deploying *Mertk* [-/-V1] mice, there is an increasing contemporary interest in the role of this RTK in what are likely to be complex traits, such as anti-tumor response and neurodegeneration (*Cook et al., 2013*; *Crittenden et al., 2016*; *Davra et al., 2021*; *Fourgeaud et al., 2016*; *Huang et al., 2021*; *Ji et al., 2013*; *Lindsay et al., 2021*; *Stanford et al., 2014*; *Tormoen et al., 2020*). There is an emerging concept of MERTK as an innate immune checkpoint and that targeting this RTK might enhance immunotherapy (*Akalu et al., 2017*; *Huelse et al., 2020*; *Lentz et al., 2021*; *Liu et al., 2021*; *Rothlin and Ghosh, 2020*). Remarkable anti-tumor resistance was observed in a number of standard mouse tumor models such as MMTV-PyVmT, B16:F10, MC38, E0771, and B78ChOva in *Mertk* [-/- V1] mice (*Cook et al., 2013*; *Davra et al., 2021*; *Lindsay et al., 2021*). Our initial results were entirely consistent with and extended such findings in that we were able to observe an astonishing anti-tumor resistance in hard-to-treat mouse models such as the anti-CTLA-4- and anti-PD-1-refractory YUMM1.7 mouse melanoma. *Mertk* [-/- V1] mice could even be rendered resistant to orthotopic GL261 glioblastoma with post-tumor DC-Vax treatment. Although our use of additional tumor models further validated the paradigm of host anti-tumor resistance in the *Mertk* [-/-V1] mice, we surprisingly also discovered that *Mertk* ablation is not sufficient for resistance against YUMM1.7 and GL261. It is important to acknowledge that the work by Davra et al. and Lindsay et al. not only employed the *Mertk* [-/-V1] mouse, but also utilized alternative approaches such as antibodies or small-molecule inhibitors to pharmacologically disable MERTK (*Davra et al., 2021*; *Lindsay et al., 2021*). The role of *Mertk* as an oncogene confounds interpretations of ostensible immune function when using pharmacological agents. It is possible that some of the reported effects are a consequence of inhibiting the oncogenic role of TAM RTKs. Since we did not test identical tumor models as the previous reports, it remains entirely possible that anti-tumor immunity in the models reported before, serendipitously, is dependent exclusively on the loss of MERTK. For instance, macrophage-specific ablation of *Mertk* in B6 background conferred a statistically significant reduction in tumor growth in the PyMT mouse model (*Sekar et al., 2022*). Taken together, our findings indicate that the universality of improving anti-tumor immunity through targeting MERTK, irrespective of the tumor model employed, has to be questioned. MERTK inhibition-dependent enhancement of anti-tumor immunity is likely to be contextual.

Similar concerns extend to *Mertk* [-/-V1] mouse traits beyond anti-tumor immunity. The conditional ablation of *Mertk* in microglia led to decreased phagocytosis index in the mouse hippocampus (*Fourgeaud et al., 2016*). Hippocampal neurogenesis was impaired in *Mertk* [-/-V1] mice (*Ji et al., 2013*) or paradoxically, transiently increased in the absence of *Mertk* (*Fourgeaud et al., 2016*), suggesting laboratory-specific strain differences in *Mertk* [-/-V1] mice. It remains to be determined if changes in hippocampal neurogenesis in *Mertk* [-/-V1] mice are a direct consequence of loss of *Mertk* or due to strain-specific expression of modifiers. The unequivocal demonstration of independent modifiers of *Mertk* [-/-V1] mouse traits calls for re-examination of the molecular and functional basis of phenotypes assigned exclusively to the loss of MERTK. We do not, however, consider all *Mertk* [-/-V1] mouse traits to be ambiguous. Using *Mertk* [-/-V1] mice, DeBerge et al. demonstrated that post-reperfusion MERTK-dependent phagocytosis is cardioprotective in a mouse model of myocardial ischemia reperfusion

injury (*DeBerge et al., 2017*). Notably, conditional ablation of *Mertk* in myeloid cells of B6 background confirmed the cardioprotective role for this RTK (*DeBerge et al., 2017*).

Neither are all phenotypes of *Mertk* [-/-V1] mice invariably and inviolably linked to loss of phagocytosis in the absence of MERTK. The functional role of MERTK in phagocytosis remains beyond doubt, and we, in fact, validated the dependency of macrophages on MERTK for phagocytosis by using two independently generated mouse knockout lines. Our studies are also consistent with a report by Wanke et al. In mice wherein MERTK signaling was disabled by targeting the sequence coding for the kinase domain of MERTK (*Mertk* [K614M/K614M] mice), peritoneal macrophages failed to phagocytose apoptotic thymocytes at rates similar to that observed in *Mertk* [-/- V1] mice (*Wanke et al., 2021*). Importantly, *Mertk* [K614M/K614M] mice were generated on a B6 background, further indicating that the phagocytic function of macrophages depended on MERTK and is lost in its absence, independent of the 129P2 or B6 ES cell background that *Mertk* was targeted in. The biological basis for the differences in retinal degeneration phenotype in the *Mertk* [-/-V1] versus *Mertk* [-/-V2] and *Mertk* [-/-V3] mouse lines can also be trivially explained by a redundancy in phagocytosis provided for by TYRO3 in the absence of MERTK. The original *Mertk* [-/-] mouse line, even after >10 generations of backcrossing to B6 mice in our laboratory, retained an ~15 cM chromosome 2 segment with 129P2 alleles that are genetically linked to the targeted *Mertk* locus and failed to recombine and segregate from it. A previous report by Vollrath et al. had also reported the presence of an ~40 cM chromosome 2 segment with 129P2 alleles in *Mertk* [-/-V1] mice (*Vollrath et al., 2015*). Vollrath et al. mapped a modifier of the retinal degeneration phenotype to a 2 cM region containing 53 ORFs by genetic linkage mapping (*Vollrath et al., 2015*). Segregation of the *Mertk*-linked 129 region with a B6 region that contained *Tyro3* restored retinal homeostasis (*Vollrath et al., 2015*). Since a genetic strategy was used in this previous report, in theory, potential contribution of other genes within the co-segregating region with suppressor function cannot be ruled out. Given that *Tyro3* is the only *Mertk* paralog within this region, such an exception is unlikely. Nonetheless, in a complementary approach, we directly targeted *Tyro3* in B6 *Mertk* [-/-V2] mice-derived ES cells to unambiguously demonstrate that the simultaneous ablation of *Mertk* and *Tyro3* in B6 mice is necessary and sufficient for retinal degeneration. One of the potential concerns for using MERTK inhibitors in the clinic is the on-target adverse event of rapid retinal degeneration. *Vollrath et al., 2015*, *Maddox et al., 2011*, and our results collectively indicate that sparing TYRO3 activity can greatly ameliorate this concern while targeting the oncogenic activity of MERTK.

The phagocytosis paradigm, nevertheless, failed entirely in explaining the remarkable anti-tumor resistance of *Mertk* [-/-V1] mice. Loss of/reduced phagocytosis by macrophages and other phagocytes, when *Mertk* is genetically ablated, is often surmised causal to the manifested phenotype (*Davra et al., 2021*; *Huang et al., 2021*; *Stanford et al., 2014*; *Zhou et al., 2020*). *Tyro3* is not expressed in BMDMs, was not a DEG in BMDM RNAseq experiments, and *Mertk* [-/-V1], *Mertk* [-/-V2], and *Mertk* [-/-V3] BMDMs were equally deficient in efferocytosis in an ex vivo assay. Thus, neither TYRO3 nor any other DEG provided a redundancy to macrophage phagocytosis in the absence of MERTK. This observation seriously challenges the dogma that failure of MERTK-dependent efferocytosis of tumor cells universally improves anti-tumor immunity.

Phenotypic differences not accounted for fully by the target gene are not entirely an uncommon occurrence in tumor resistance. For example, the frequency of intestinal adenomas in *Apc* [min] mice is dramatically modified by the genetic background that the mutation is introduced in. *Apc* [min] mice with a B6 background were reported to develop 28.5 ± 7.9 tumors and die within 4 months of birth (*Dietrich et al., 1993*). When these *Apc* [min] mice were crossed to an AKR mouse, the mixed B6/AKR background had 5.8 ± 4.3 tumors and lived until they were euthanized at 300 days (*Dietrich et al., 1993*). Crossing the B6/AKR F1 mice with AKR further reduced tumor loads to 1.75 ± 1.7 tumors (*Dietrich et al., 1993*). The modifier allele was identified as *Mom1* mapping to chromosome 4 (*Dietrich et al., 1993*). We postulate that the anti-tumor resistance of the *Mertk* [-/-V1] mouse line also involves modifier alleles. The 129P2 segment in *Mertk* [-/-V1] mice not only accounts for a number of gene expression differences within the *cis* non-B6 flanking region, but also results in differential gene expression in a number of *trans* loci. It is possible that the anti-tumor resistance ascribed solely to *Mertk* might

actually involve one or more modifier activities encoded within *cis* or *trans* loci unique to the *Mertk* [-/-V1] mice that are absent in B6 ES cell-derived *Mertk* [-/-] mouse lines. Furthermore, since gene expression differences between *Mertk* [-/-V1] and *Mertk* [-/-V2] or *Mertk* [-/- V3] mice are cell type-specific, it is possible that tissue-specific expression quantitative trait loci (eQTLs) may function in combination as modifiers of complex traits, such as anti-tumor immune responses. This makes identification of the crucial modifiers more challenging. Assuming said modifiers are acting in *cis*, genomic CRISPR/CAS9 screens for loss of YUMM1.7 and GL261 resistance in a series of *Mertk* [-/-V1]-derived mouse lines ablated for genes within the ~15 cM region of chromosome 2 may reveal the modifiers. Alternatively, if the 129P2 allele is dominant, BAC transgenics using *Mertk* [-/-V2] ES cells to express 129P2 variants of genes for the rescue of the anti-tumor response might also be an appropriate strategy.

We cannot also entirely eliminate the possibility that *Mertk* itself does not have a functional role in some of the traits observed in *Mertk* [-/-V1] mice. For example, the DC-intrinsic defect in emigration to inflamed tissue that was initially ascribed to loss of function of *Nlrp10* in *Nlrp10* [-/-] mice first generated in a B6-BALB/c mixed background was subsequently revealed to be caused by a spontaneous mutation in *Dock8*, an unexpected by-product of the BALB/c variant of the gene (*Eisenbarth et al., 2012*; *Krishnaswamy et al., 2015*). Similarly, *Il10* [-/-] mice in a B6 background differed in open-field behavioral tests from *Il10* [-/-] mice in a 129-B6 mixed background (*Bolivar et al., 2001*). The behavioral differences were found to be due to eQTLs *Emo4* and *Reb1* inherited from flanking region of *Il10* in chromosome 1 and not attributable to *Il10* itself (*de Ledesma et al., 2006*).

In conclusion, two newly generated mouse lines with genetic ablation of *Mertk* reveal that some of the well-established *Mertk* [-/-V1] mice traits tested herein were products of epistatic interactions with modifiers in the 129P2 genome. These new *Mertk* knockout mouse lines should prove useful for validation of phenotypes ascribed exclusively to *Mertk* based on studies employing *Mertk* [-/-V1] mice. Furthermore, these studies also identify a unique anti-tumor resistance in *Mertk* [-/- V1] mice against anti-CTLA-4 and anti-PD-1 refractory YUMM1.7 melanoma as well as against GL261 brain tumors, the molecular basis of which remains unknown. This model may yet prove useful for the discovery of a novel host anti-tumor response that can be therapeutically harnessed for improving outcomes in cancer.

# Materials and methods

## Key resources table

| Reagent type (species) or resource | Designation | Source or reference | Identifiers | Additional information |
|---|---|---|---|---|
| Gene (*Mus musculus*) | *Mertk* | NCBI Gene ID | 17289 | |
| Gene (*M. musculus*) | *Tyro3* | NCBI Gene ID | 22174 | |
| Strain, strain background (*M. musculus*, males and females) | C57BL/6J | The Jackson Laboratory | Strain# 000664 | |
| Genetic reagent (*M. musculus*) | *Mertk* [-/-V1] | *Lu et al., 1999* | | Both males and females used |
| Genetic reagent (*M. musculus*) | *Mertk* [-/-V2] | This paper | | See section 'Animals', both males and females used |
| Genetic reagent (*M. musculus*) | *Mertk* [-/-V3] | This paper | | See section 'Animals', both males and females used |
| Genetic reagent (*M. musculus*) | *Tyro3* [-/-] | *Lu et al., 1999* | | Both males and females used |
| Genetic reagent (*M. musculus*) | *Mertk* [-/-V2] *Tyro3* [-/-V2] | This paper | | See section 'Animals', both males and females used |
| Biological sample (*M. musculus*) | Retinal pigment epithelial cells | This paper | | Freshly isolated from indicated mouse strains, see 'Materials and methods' sections for various applications |
| Biological sample (*M. musculus*) | Bone marrow-derived macrophages | This paper | | Freshly isolated from indicated mouse strains, see 'Materials and methods' sections for various applications |

*Continued on next page*

*Continued*

| Reagent type (species) or resource | Designation | Source or reference | Identifiers | Additional information |
|---|---|---|---|---|
| Antibody | Anti-MERTK (rabbit polyclonal) | Abcam | Cat# ab95925 | WB (1:1000) |
| Antibody | Anti-TYRO3 (rabbit monoclonal) | Cell Signaling Technology | Cat# 5585S | WB (1:1000) |
| Antibody | Anti-β-actin (rabbit monoclonal) | Cell Signaling Technology | Cat# 8457S | WB (1:10,000) |
| Antibody | Anti-β-actin (mouse monoclonal) | Cell Signaling Technology | Cat# 3700S | WB (1:10,000) |
| Antibody | Anti-rabbit IgG, IRDye800 (donkey polyclonal) | Li-COR Biosciences | Cat# 926-32213 | WB (1:15,000) |
| Antibody | Anti-mouse IgG, IRDye680 (donkey polyclonal) | Li-COR Biosciences | Cat# 926-68072 | WB (1:20,000) |
| Antibody | Anti-mouse CD16/32 (rat monoclonal) | BioLegend | Cat# 101302 | FC (1:1000) |
| Antibody | Anti-mouse CD11b (rat monoclonal) | BioLegend | Cat# 101222 | FC (1:200) |
| Antibody | Anti-mouse F4/80 (rat monoclonal) | BioLegend | Cat# 123114 | FC (1:200) |
| Antibody | Anti-mouse MERTK (rat monoclonal) | Invitrogen | Cat# 17-5751-82 | FC (1:200) |
| Chemical compound, drug | RNAprotect Cell Reagent | QIAGEN | Cat# 76526 | |
| Chemical compound, drug | Paraformaldehyde 16% | Electron Microscopy Sciences | Cat# 15710 | |
| Chemical compound, drug | Glutaraldehyde 25% | Electron Microscopy Sciences | Cat# 16200 | |
| Chemical compound, drug | Tropicamide ophthalmic Solution 0.5% | Sandoz | Cat# 61214-354-01 | |
| Chemical compound, drug | cOmplete Protease Inhibitor Cocktail | Roche | Cat# 11697498001 | |
| Chemical compound, drug | Eye Fixative | ServiceBio | Cat# G1109 | |
| Commercial assay, kit | RNeasy Plus Micro Kit | QIAGEN | Cat# 74034 | |
| Commercial assay, kit | iScript cDNA Synthesis Kit | Bio-Rad | Cat# 1708891 | |
| Commercial assay, kit | Mini-PROTEAN TGX Stain-Free Precast Gel | Bio-Rad | Cat# 4568025, 4568125 | |
| Commercial assay, kit | Immun-Blot PVDF membrane | Bio-Rad | Cat#1620177 | |
| Commercial assay, kit | Pierce BCA Protein Assay Kit | Thermo Scientific | Cat# 23227 | |
| Other | Hematoxylin-eosin staining | iHisto histopatholgy support | | https://www.ihisto.io/ |

## Animals

Animals were bred and maintained under a strict 12 hr light cycle and fed with standard chow diet in a specific pathogen-free facility at Yale University. All experiments involving animals were performed in accordance with regulatory guidelines and standards set by the Institutional Animal Care and Use Committee of Yale University. All C57BL/6 mice were purchased from Jackson Laboratories and subsequently bred and housed at Yale University. The widely used *Mertk* [-/-V1] mice have been described previously (*Camenisch et al., 1999*; *Lu et al., 1999*). *Mertk* [-/-V1] mice were crossed onto the C57BL/6J background (Jackson Laboratories strain# 000664) for at least 10 generations. As shown in *Figure 1*, *Mertk* [-/-V1] mice have deletion of exon 17 that corresponds to the kinase domain of *Mertk* and the

neomycin cassette is still present in the *Mertk* locus. Of note, exon 17 was referred to as exon 18 in the original description of the mouse (*Lu et al., 1999*).

To generate an independent line of mice that has *Mertk* deleted globally (referred to as *Mertk* -/-V2 mice), we bred *Mertk* f/f mice with commercially available Rosa26ERT2Cre+ (Jackson Laboratories strain# 008463) mice. A description detailing the generation of the *Mertk* f/f mice can be found in *Fourgeaud et al., 2016* (*Fourgeaud et al., 2016*). *Mertk* f/f mice originated from embryos of C57BL/6NJ background. Germline inactivation of the *Mertk* f/f allele in *Mertk* f/f Rosa26ERT2Cre+ mice was achieved by intraperitoneally injecting 3 mg of 4-hydroxytamoxifen (Sigma-Aldrich H7904) for five consecutive days. Two weeks post-tamoxifen injection, adult males and females were set up as breeder pairs. Litters from this cross were then screened for excision of the *Mertk* f/f allele. Once identified, excision positive mice were bred with C57BL/6J mice to eliminate the Cre recombinase. Finally, mice were genotyped to set apart *Mertk* -/-V2 founder mice that had excised exon 18, encoding for the kinase domain of *Mertk*, on both alleles. Genome-wide single-nucleotide polymorphism (SNP) analysis of our established *Mertk* -/-V2 mice indicated that ~84.37% of their genome was C57BL/6J-derived while ~15.62% was C57BL/6NJ-derived. An independent line targeting deletion of *Mertk,* designated as *Mertk* -/-V3 mice, was generated at Cyagen Biosciences Inc (Santa Clara, CA), by CRISPR/Cas9-mediated genome engineering. As shown in *Figure 1*, *Mertk* -/-V3 mice have exons 3 and 4 targeted; transcribed mRNA from targeted allele with frameshift mutation undergo nonsense-mediated decay. Single-guide RNAs (sgRNAs) were injected into fertilized C57BL/6NJ eggs, and founder animals were identified by PCR followed by DNA sequencing analysis. Genome-wide SNP analysis revealed that ~55.22% of their genome was C57BL/6J-derived and ~44.73% was C57BL/6NJ-derived.

CRISPR/Cas9 technique was used to make *Mertk* -/-V2 *Tyro3* -/-V2 mice, as described previously (*Henao-Mejia et al., 2016*), at the Yale Genome Editing Center. In brief, T7-sgRNA templates were prepared by PCR, incorporating the guide sequences from the desired target regions in the mouse *Tyro3* gene (NCBI Gene ID: 22174), with a 5′ guide sequence of CTACACCTACAGAGAACAAG (sense orientation, cutting the gene at 8387/8 bp within intron 6) and a 3′ guide sequence of CCCAAGTG TCAGAATCCCAG (sense orientation, cutting the gene at 17,800/1 bp within intron 18), thus resulting in a deletion of 9413 bp. The T7-sgRNA PCR templates were then used for in vitro transcription and purification with the MEGAshortscript T7 Transcription Kit and MEGAclear Transcription Clean-Up Kit, respectively (both from Thermo Fisher Scientific AM1354, AM1908). Cas9 mRNA (CleanCap, 5-methoxyuridine-modified) was purchased from TriLink Biotechnologies. Subsequently, cytoplasmic microinjections of sgRNAs and Cas9 mRNA into single-cell embryos obtained from *Mertk* -/-V2 donors were performed. Founder mice with heterozygous deletion of the *Tyro3* allele were identified with genotyping by PCR. Next, *Mertk* -/-V2 *Tyro3* -/-V2 line was established by heterozygote-to-heterozygote breeding. Consistent with previous reports (*Chen et al., 2009*; *Lu et al., 1999*), male double knockout mice lacking both *Mertk* and *Tyro3* have significantly smaller testicles and reduced fertility. *Tyro3* -/-V1 mice have been described previously (*Lu et al., 1999*).

## PCR amplification

PCR reactions for sequencing were performed using the primers listed in *Appendix 1—table 1*. PCR amplifications were carried out with TopTaq Master Mix Kit (QIAGEN). PCR reactions of 25 µl were performed with 2 µl genomic DNA, 0.2 µM primer pair, 2.5 µl CoralLoad Concentrate 10×, 12.5 µl TopTaq Master Mix, 2× (contains TopTaq DNA polymerase, TopTaq PCR Buffer with 3 mM MgCl₂ and 400 µM each dNTP). Thermal cycling conditions were based on Touchdown PCR method described by *Korbie and Mattick, 2008*. PCR products were examined by gel electrophoresis.

## STR analysis

Genomic DNA was isolated from liver biopsies using the DNeasy blood and tissue kit (QIAGEN 69504) according to the manufacturer's protocol. PCR reactions to amplify 24 different microsatellite regions on chromosome 2 were performed using the primers listed in *Appendix 1—table 1*. Thermal cycling conditions were optimized based on Touchdown PCR method described by *Korbie and Mattick, 2008*. Following PCR amplification, STR markers were scored by resolving PCR products on 4% agarose gels.

## Western blot

Protein lysates from adult mice RPE were obtained using a previously validated protocol (*Wei et al., 2016*). Briefly, the neural retina was removed and posterior eyecups were incubated on ice in 200 µl of RIPA buffer with protease inhibitor cocktail, EDTA-free (Roche 11697498001) for up to 1 hr. Posterior eyecups were removed and the dislodged RPE cells were sonicated for 20 s on ice. After 10 min at 14,000 rpm in a refrigerated centrifuge, supernatants were transferred to new tubes and protein content was quantified with Pierce BCA assay (Thermo Fisher Scientific 23227) as per the manufacturer's instructions. Concomitantly, spleen, brain, testes, and peritoneal exudate were collected from adult mice. Samples were kept in NP-40 buffer containing a cocktail of protease inhibitors (Roche 11697498001). Tissues were mechanically disrupted and left rotating at 4°C for 2 hr to ensure complete homogenization. Subsequently, all samples were centrifuged for 20 min at 12,000 rpm and the supernatants were collected for protein quantification as described above.

For immunoblots, equal amounts of total protein in Laemmli buffer were subjected to electrophoresis on precast polyacrylamide gels (Bio-Rad 4568025, 4568125) and transferred to PVDF membranes (Bio-Rad 1620177). Membranes were blocked and probed overnight with corresponding primary antibodies (MERTK: Abcam ab95925 1/1000, TYRO3: Cell Signaling Technology 5585S 1/1000, β-actin: Cell Signaling Technology 8457S and 3700S). Secondary antibodies conjugated to near-infrared fluorophores (LI-COR Biosciences 926-32213, 926-68072) were detected using Odyssey Classic Imaging System (LI-COR Biosciences) and quantified with Image Studio Lite Software (LI-COR Biosciences).

## Generation of BMDMs

BMDMs from age-matched, adult C57BL/6 *Mertk* $^{-/-V1}$, *Mertk* $^{-/-V2}$, and *Mertk* $^{-/-V3}$ mice were differentiated from bone marrow precursors. Briefly, bone marrow cells were isolated and propagated for 7 days in 30% L929-conditioned RPMI (Gibco 11875101) containing 20% FCS (Sigma-Aldrich 18D078) and 1% Penicillin-Streptomycin (Gibco 10378016). At day 7, BMDMs were lifted for downstream assays.

## Phagocytosis assays

Thymocytes were isolated from 3- to 6-week-old C57BL/6 mice. Thymocytes were incubated for 4 hr with 1 µg/ml of dexamethasone (Sigma-Aldrich D4902) to induce apoptosis. In parallel, BMDMs from the indicated mice were collected and replated to adhere for 3 hr at 37°C. Apoptotic thymocytes, pre-labeled with 0.1 mg/ml of pHrodo-SE (Thermo Fisher P36600), were subsequently added to BMDMs at 6:1 ratio. Cells were co-cultured for 1 hr at 37°C. Afterward, the apoptotic thymocyte-containing media were removed and BMDMs were washed five times with 1× PBS. Adherent BMDMs were then treated with Accutase (Sigma-Aldrich A6964) for 10 min at 37°C. BMDMs were then gently scraped off for collection and stained for analysis with flow cytometry.

## Flow cytometry staining and acquisition

Single-cell suspensions of BMDMs were stained in PBS with fixable viability dye (Thermo Fisher Scientific 65-0865-14) for 10 min. Cells were then incubated in 2% FCS/PBS solution containing anti-mouse CD16/32 antibody (BioLegend clone 93) for 15 min and subsequently stained with a combination of fluorophore-conjugated primary antibodies against mouse CD11b (BioLegend clone M1/70), F4/80 (BioLegend clone BM8), and MERTK (Invitrogen clone DS5MMER) at 4°C for 25 min. After staining, cells were washed and data was immediately acquired with BD LSRII flow cytometer using BD FACS-Diva software (BD Biosciences). Finally, raw data were analyzed using FlowJo software (Tree Star Inc).

## Histological analysis

After mice were sacrificed by carbon dioxide inhalation, eyecups were immediately collected and incubated overnight in eye fixative (ServiceBio). Hematoxylin and eosin staining was performed by iHisto Inc Samples were processed, embedded in paraffin, and sectioned at 4 µm. Paraffin sections were then deparaffinized and hydrated using the following steps: xylene, two rounds,15 min each; 100% ethanol, two rounds, 5 min each; 75% ethanol, one round, 5 min; and 1× PBS, three rounds, 5 min each at room temperature. After deparaffinization, 4-µm-sectioned samples were placed on glass slides and stained with hematoxylin and eosin. Whole-slide scanning (20×) was performed on an EasyScan Infinity (Motic). ONL thickness was analyzed in ImageJ (NIH). Quantification of ONL

thickness was performed in the medial retina. The areas analyzed were defined by distance from the optic disk. Ten measurements of ONL thickness were done per mouse.

## Electron microscopy

Adult mice were perfused with 1× PBS followed by 4% paraformaldehyde (Electron Microscopy Sciences 15710) in PBS. Eyeballs were then carefully dissected and further fixed in 2.5% glutaraldehyde (Electron Microscopy Sciences 16200) and 2% paraformaldehyde in 0.1 M sodium cacodylate buffer (pH 7.4) for 1 hr at room temperature. Next, eyeballs were post-fixed in 1% $OsO_4$ for 1 hr at room temperature and *en bloc* stained with 2% aqueous uranyl acetate for 30 min. They were then dehydrated in a graded series of ethanol, going from 70% to 100%, and finally transferred to 100% propylene oxide before being embedded in EMbed 812 resin, polymerized at 60°C overnight. Samples from medial retinas were cut respectively into thin sections of 60 nm by a Leica ultramicrotome (UC7), placed on standard EM grids, and stained with 2% uranyl acetate and lead citrate. Retinal samples were examined with a FEI Tecnai transmission electron microscope at 80 kV accelerating voltage, and digital images were recorded with an Olympus Morada CCD camera and iTEM imaging software at the Yale Center for Cellular and Molecular Imaging (CCMI) Electron Microscopy Facility.

## ERG recordings

All experimental animals were adapted in a dark room for 12 hr prior to recordings. Animals were anesthetized under dim red illumination using a 100 mg/kg ketamine and 10 mg/kg xylazine cocktail injected intraperitoneally and pupils were dilated by application of a 0.5% tropicamide eye drop (Sandoz 61214-354-01) at least 15 min before recordings. The cornea was intermittently irrigated with balanced salt solution to maintain the baseline recording and prevent keratopathy. Scotopic electroretinograms were acquired with UTAS ERG System with a BigShot Ganzfeld Stimulator (LKC Technologies, Inc). A needle reference electrode was placed under the skin of the back of the head, a ground electrode was attached subcutaneously to the tail, and a lens electrode was placed in contact with the central cornea. The scotopic response was recorded for different luminances (i.e., $log_2$ −2.1, −0.6, 0.4, and 1.4 cd.s/m$^2$) using EMWin software, following the manufacturer's instructions (LKC Technologies, Inc). The a-wave was measured as the difference in amplitude between baseline recording and the trough of the negative deflection, and the b-wave amplitude was measured from the trough of the a-wave to the peak of the ERG.

## Tumor implantations

A total of 100,000 YUMM1.7 melanoma cells, resuspended in 50 ul of sterile 1× PBS, were subcutaneously injected into shaved rear flank of 6- to 12-week-old male mice. Mice were monitored for tumor growth by measuring the length and width of tumor masses using a caliper. Tumor volumes were scored with the formula $(A \times B^2) \times 0.4$, in which A is the largest and B is the shortest dimension. Each mouse was said to have reached the end of its tumor-free survival when the largest dimension of its tumor was measured to be 5 mm. Mice were sacrificed once tumor growth reached an endpoint cutoff of 1000 mm$^3$.

Anesthetized 8-to-12-week-old male mice were placed in a stereotactic apparatus and an incision was made with a scalpel over the cranial midline. A burr hole was made 1 mm lateral and 2 mm anterior to the bregma. A needle containing a suspension of GL261-luciferase (Luc) cells was inserted to a depth of 3 mm. After the needle is allowed to rest in the burr hole for 5 min, 10,000 GL261-Luc cells were infused over the course of 4 min. Once cells were injected, the needle was allowed to rest in the skull for 5 min before removal. Finally, the incision was closed with vetbond tissue adhesive and animals were administered the full course of postoperative analgesic drugs, according to regulatory guidelines and standards set by the Institutional Animal Care and Use Committee of Yale University. Intracranial tumor growth was monitored using the 3D image reconstruction feature of the IVIS Spectrum instrument. Mice received an intraperitoneal injection of 150 mg/kg of luciferin (Goldbio LUCK) prior to imaging. Tumor-bearing mice were checked daily for clinical signs of sickness behavior and were euthanized when one or more of the following symptoms were present: hunching, decreased activity, head tilt, weight loss, seizures, and failure to groom.

## Generation of BMDCs

BMDCs were differentiated from granulocyte macrophage colony-stimulating factor (GM-CSF) as previously described (*Inaba et al., 1992*). Briefly, bone marrow progenitors were collected from the femurs of adult C57BL/6 WT mice and $10 \times 10^6$ progenitor cells were cultured in RPMI media (Gibco 11875101) supplemented with 10% FBS (Sigma-Aldrich 18D078), 1% Penicillin-Streptomycin (Gibco 10378016), and 20 ng/ml of recombinant murine GM-CSF (PeproTech 315-03). On days 3 and 5, more supplemented media were added and cells were left in culture until day 7 when they would be ready to be used for generation of DC-Vax, as described below.

## Preparation of dendritic cell vaccines

GL261 tumor cell lysates were made by subjecting cells to six rounds of rapid freeze–thaw cycles, comprised of 3 min of incubation in liquid nitrogen and 4 min of incubation at 56°C. BMDCs were then incubated with GL261 tumor lysate (1 mg of lysate/$10 \times 10^6$ BMDCs) for 2 hr at 37°C. After 2 hr, 1 ug/ml LPS (Sigma L2630) was added to BMDC-tumor lysate suspension and cells were incubated for 24 hr at 37°C. Next, supernatant was aspirated, and BMDCs were collected and washed with 1× PBS three times. Finally, $1 \times 10^6$ GL261-lysate pulsed BMDCs were intraperitoneally injected into mice at days 14 and 21 post-intracranial implantation of GL261-Luc cells, as detailed above.

## Total RNA isolation and sequencing analysis

RNA was collected from postnatal day 25 mice using a previously validated method (*Xin-Zhao Wang et al., 2012*). Briefly, after euthanasia, mice eyes were enucleated and the posterior eyecup was incubated on ice in 400 µl of RNAprotect (QIAGEN 76526) for 1 hr. RPE-containing tubes were agitated for 10 min to dislodge any RPE cells attached to the posterior eyecup and centrifuged for 5 min at 685 × *g*. The RPE pellet was then subjected to total RNA extraction using RNeasy Mini kit (QIAGEN 74106) following the manufacturer's instructions. Similarly, total RNA from BMDMs was extracted from these cells using RNeasy Mini kit (QIAGEN 74106) following the manufacturer's instructions.

RNA libraries from BMDMs and RPE cells were prepared at the Yale Keck Biotechnology Resource Laboratory from three to six biological replicates per condition. Samples were sequenced using 150 bp base pair paired-end reading on a NovaSeq 6000 instrument (Illumina). The raw reads were then subjected to trimming by btrim (*Kong, 2011*) to remove sequencing adaptors and low-quality regions. Next, reads were mapped to the mouse genome (GRCm38) using STAR (*Dobin et al., 2013*). Finally, the Deseq2 (*Love et al., 2014*) package was run to identify DEGs according to the p-values adjusted for multiple comparisons. Genes with p-adjusted values <0.05 and $\log_2$ fold change ≤−1.25 or ≥1.25 were considered differentially expressed. All data RNA-sequencing datasets and the processed data that support the findings of this study have been deposited to the Gene Expression Omnibus (GEO) under accession ID: GSE205070.

## Quantitative PCR analysis

Reverse transcription of RNA was performed utilizing iScript cDNA Synthesis Kit (Bio-Rad 1708891). Using KAPA SYBR Fast qPCR Kit (Kapa Biosystems KK4602), we amplified cDNA fragments and proceeded with qPCR reactions on CFX96 Thermal Cycler Real Time System (Bio-Rad). The reactions were normalized to three housekeeping genes (*Gapdh, Hprt,* and *Rn18s*), and specificity of the amplified products was verified by looking at the dissociation curves. All oligonucleotides for qPCR were either purchased from Sigma-Aldrich or produced at the Yale University Keck Oligonucleotide Synthesis Facility (see sequences in *Appendix 1—table 1*).

## Statistical analysis

All statistical analyses were done using GraphPad Prism (GraphPad Software Inc). All data are shown as mean ± SEM, and each data point represents a unique animal. Statistical differences between experimental groups were determined by employing various tests, namely, Kaplan–Meier test, Mann–Whitney test, Student's *t*-test, one-way and two-way ANOVAs. Additionally, hypergeometric distribution analysis was employed to determine which chromosomes are over-represented in the pool of DEGs associated with each genotype. The distribution of DEGs was said to be enriched on a chromosome when the probability of association with a chromosome had p-value <0.05 compared to the number of DEGs that would be expected to map to each chromosome by chance.

## Data availability

All mice described are available upon request from the Rothlin Ghosh laboratory. Requests should be directed to carla.rothlin@yale.edu and sourav.ghosh@yale.edu. RNA-sequencing datasets and the processed data that support the findings of this study have been deposited to the GEO under accession ID: GSE205070. All data generated or analyzed during this study are included in the article and supporting files.

## Acknowledgements

The authors acknowledge the members of the Rothlin-Ghosh laboratory for scientific discussions relating to the preparation of this manuscript. The authors also thank Dr Sagie Wagage for technical contributions with GL261 model experiments and Ms Veronica Galimberti for genotyping. This study was funded by NIH R01CA212376 (CVR and SG), Howard Hughes Medical Institute Faculty Scholar award (CVR) and a YSPORE Career Development Award DRP27 (CVR). SCF is supported by the Kim B and Stephen E Bepler Professorship in Biology. MA is the recipient of a long-term fellowship (BUIT 2012-5347) from the Dutch Cancer Society. LDH was awarded NSF DGE-1122492 and Richard K Gershon Fellowship. Research reported in this publication was supported by the National Cancer Institute of the National Institutes of Health under award number 2T32CA193200-06 (PIs PG and DFS).

## Additional information

### Competing interests

Li-Zhen He, Diego Alvarado, Tibor Keler: is affiliated with Celldex Therapeutics. The author has no financial interests to declare. Carla V Rothlin: Senior editor, eLife. Sourav Ghosh: has received grant support from Mirati Therapeutics. The other authors declare that no competing interests exist.

### Funding

| Funder | Grant reference number | Author |
|---|---|---|
| National Institutes of Health | R01CA212376 | Carla V Rothlin |
| Howard Hughes Medical Institute | | Carla V Rothlin |
| Yale Cancer Center | YSPORE Career Development Award DRP27 | Carla V Rothlin |
| Fordham University | Kim B. and Stephen E. Bepler Professorship in Biology | Silvia C Finnemann |
| Dutch Cancer Society | BUIT 2012-5347 | Marleen Ansems |
| National Science Foundation | DGE-1122492 | Lindsey D Hughes |
| Yale University | Richard K. Gershon Fellowship | Lindsey D Hughes |
| National Cancer Institute | 2T32CA193200-06 | James Nevin |

The funders had no role in study design, data collection and interpretation, or the decision to submit the work for publication.

### Author contributions

Yemsratch T Akalu, Maria E Mercau, Data curation, Formal analysis, Validation, Investigation, Visualization, Writing – original draft, Writing – review and editing; Marleen Ansems, Formal analysis, Investigation; Lindsey D Hughes, Formal analysis, Investigation, Writing – review and editing; James Nevin, Writing – review and editing; Emily J Alberto, Xinran N Liu, Investigation; Li-Zhen He, Diego Alvarado, Tibor Keler, Resources, Methodology; Yong Kong, Formal analysis; William M Philbrick,

Methodology; Marcus Bosenberg, Resources; Silvia C Finnemann, Antonio Iavarone, Anna Lasorella, Conceptualization, Writing – review and editing; Carla V Rothlin, Sourav Ghosh, Conceptualization, Resources, Supervision, Funding acquisition, Visualization, Writing – original draft, Project administration, Writing – review and editing

### Author ORCIDs
Yemsratch T Akalu ⓘ http://orcid.org/0000-0003-0829-7972
Maria E Mercau ⓘ http://orcid.org/0000-0001-6971-8676
Lindsey D Hughes ⓘ http://orcid.org/0000-0002-1764-4553
Yong Kong ⓘ http://orcid.org/0000-0002-2881-5274
Silvia C Finnemann ⓘ http://orcid.org/0000-0001-9298-0736
Carla V Rothlin ⓘ http://orcid.org/0000-0002-5693-5572
Sourav Ghosh ⓘ http://orcid.org/0000-0001-5990-8708

### Ethics
All experiments involving animals were performed in accordance with regulatory guidelines and standards set by the Institutional Animal Care and Use Committee (IACUC) protocol (#2021-11312) of Yale University.

### Decision letter and Author response
Decision letter https://doi.org/10.7554/eLife.80530.sa1
Author response https://doi.org/10.7554/eLife.80530.sa2

## Additional files

### Supplementary files
• MDAR checklist

### Data availability
RNA-sequencing data sets and the processed data that support the findings of this study have been deposited to the Gene Expression Omnibus (GEO) under accession ID: GSE205070. All data generated or analyzed during this study are included in the manuscript and supporting files. Source data files have been provided for all figures included.

The following dataset was generated:

| Author(s) | Year | Dataset title | Dataset URL | Database and Identifier |
|---|---|---|---|---|
| Akalu YT, Mercau ME, Ansems M, Wagage S, Hughes LD, Nevin J, Alberto E, Liu X, He L, Alvarado D, Keler T, Kong Y, Philbrick WM, Finnemann SC, Iavarone A, Lasorella A, Rothlin CV, Ghosh S | 2022 | Tissue-specific modifier alleles determine Mertk loss-of-function traits | https://www.ncbi.nlm.nih.gov/geo/query/acc.cgi?acc=GSE205070 | NCBI Gene Expression Omnibus, GSE205070 |

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

# Appendix 1

**Appendix 1—table 1.** Sequence-based reagents utilized in this article.

Single-guide (sgRNAs) and repair oligonucleotides that were used in CRISPR/Cas9-based generation of *Mertk* [-/-V2]*Tyro3* [-/-V2] mice are indicated. Primers (P1–10) utilized for PCR genotyping of mouse lines targeting *Mertk* and *Tyro3*, as shown in **Figure 1—figure supplement 1**, are included. Additionally, primers used for qPCR quantification of various genes are listed. Finally, primers used for genotyping various microsatellite markers on mouse chromosome 2 are indicated with a *D2Mit*-prefix.

| Oligonucleotide | Sequence |
| --- | --- |
| sgRNA#1 | GAAATTAATACGACTCACTATAGGGAGA**CTACACCTACAGAGAACAAG**GTTTTAGAGCTAGAAATAGCAAGTTAAAATAAGGCTAGTCCGTTATCAACTTGAAAAAGTGGCACCGAGTCGGTGCTTTTTT |
| sgRNA#2 | GAAATTAATACGACTCACTATAGGGAGA**CCCAAGTGTCAGAATCCCAG**GTTTTAGAGCTAGAAATAGCAAGTTAAAATAAGGCTAGTCCGTTATCAACTTGAAAAAGTGGCACCGAGTCGGTGCTTTTTT |
| Repair Oligo#1 | ATGAAGATCAATCACAGCTGATATTCCTCCCTCTTACATCCTGCTACTACACCTACAGAGAACAAGAGGAAGAGGAAGGCTAACAAACCCTGGCGAAGTTTGTCTGTCTGTCTGTTTGTTTGTTTG |
| Repair Oligo#2 | TTAGAGCTGAGAGGTACCCTAGACTTCAGATCCTCCTGCCACCACGCCCAAGTGTCAGAATCCCAGCGGTGTAGCACTTATGAGGTACTGGAGGTCATACCTGGGACTCTGTGCACACTGAGCAAG |
| P1 | GGGGCAGAGTACCTTGCTTT |
| P2 | CTGCGTGCAATCCATCTTGT |
| P3 | CTTTCGACCTGCAGCCAATATG |
| P4 | CCTCATCCCATATCAACACTGC |
| P5 | GCTCCAGCCCCTTTTACTTTTTGT |
| P6 | GATGTGCGATGTGATGGGAGGTAG |
| P7 | CTTAGAGACCAGGCAAGGTAGAAGCA |
| P8 | TCCTGAACACTCGCTGAATGCA |
| P9 | CAGGCCTGTGCTTTCTTTATGCTA |
| P10 | ACTGCTCTTCTGGGGGTTCTGA |
| *Tyro3* F | TGCCATTCGTACCGACTCAG |
| *Tyro3* R | TTTCTTGGACCCAGGACAGCTT |
| *Gchfr* F | CCACGCACCATGCCCTATCT |
| *Gchfr* R | GCTCCGGATCCGAGTGTTCA |
| *Gatm* F | TGCACTACATCGGCTCTCGG |
| *Gatm* R | CAGAGGATGGGTGGCCTTGT |
| *Thbs1* F | GGGGAGATAACGGTGTGTTTG |
| *Thbs1* R | CGGGGATCAGGTTGGCATT |
| *Pmepa1* F | AGCATGGAGATCACGGAGCTG |
| *Pmepa1* R | ACCGTACTCTCTGAGGGCCA |
| *Exd1* F | TCCCAGCAGTGACTACCATTT |
| *Exd1* R | CTCGTGCCCGAAGAACACTTT |
| *Itpka* F | CACGAAGCCGAGAGCAAGTG |
| *Itpka* R | TGAGCTGCCAATCACCTCGT |

*Appendix 1—table 1 Continued on next page*

*Appendix 1—table 1 Continued*

| Oligonucleotide | Sequence |
| --- | --- |
| *Duoxa2* F | CGTTAACATTACACTCCGAGGAACA |
| *Duoxa2* R | CAGAATGCCACCCACAGTGT |
| *Aa467197* F | GGAGCCACATCTTTCGCTTTG |
| *Aa467197* R | CTCCTCAACGGGCTTCCATTG |
| *Gapdh* F | TCCCACTCTTCCACCTTCGA |
| *Gapdh* R | AGTTGGGATAGGGCCTCTCTT |
| *Hprt* F | AAGCTTGCTGGTGAAAAGGA |
| *Hprt* R | TTGCGCTCATCTTAGGCTTT |
| *Rn18s* F | GTAACCCGTTGAACCCCATT |
| *Rn18s* R | CCATCCAATCGGTACTAGCG |
| *D2Mit149* F | ATATCATATAGTAGAGAAAGCGTGCTG |
| *D2Mit149* R | TCATTAGACTTGGAAAAAAGTTTGC |
| *D2Mit152* F | CACAGATCTTGTAAGACCACGTG |
| *D2Mit152* R | TGCCATGAGTGTGGGACTAA |
| *D2Mit323* F | AGAATCCTAAGTGGTGGTTAGAGG |
| *D2Mit323* R | ACCCAAAGTTGTCTTTAAGTACACA |
| *D2Mit328* F | CTTTCAATGTTCCGGCATG |
| *D2Mit328* R | AAGACTTGCTTTCATTAGACCACA |
| *D2Mit42* F | ATTACTGGGCAGGAACATTTG |
| *D2Mit42* R | GCCAAACTTCCAGACTCCTC |
| *D2Mit206* F | TGTCAGAACTGGACAATGTCG |
| *D2Mit206* R | ATGATAACAGACACTAATGATTAGGGC |
| *D2Mit101* F | ATAATTCCTGATTTGCTGTTTGTG |
| *D2Mit101* R | ACATGAAGCCTAGAGGGTGC |
| *D2Mit62* F | GGATACCGTTTGGAAAGTAAACC |
| *D2Mit62* R | GCAAGAAGCACAGGAGGC |
| *D2Mit395* F | AGGTCAGCCTGGACTATATGG |
| *D2Mit395* R | AGCATCCATGGGATAATGGT |
| *D2Mit445* F | CCTATACACGCACACACAGACA |
| *D2Mit445* R | ATGCCCTGCTTGCTATTGTT |
| *D2Mit397* F | TGATGAAGGTTCTTTTTCTCCC |
| *D2Mit397* R | CCACAGTTGGTAATTATCTGGC |
| *D2Mit164* F | TCTCTGCTAATTAAGTTGAAGAGTGC |
| *D2Mit164* R | ACCAGTGTGTGTTTGTATGATGTG |
| *D2Mit255* F | GCAAGTGTGATCTGGGTGC |
| *D2Mit255* R | TGAGCACACTTACACTGTGGTG |
| *D2Mit106* F | GAGGGTTGCCAAAGAGACTG |
| *D2Mit106* R | CACCTCAGGGGAACATTGTG |

*Appendix 1—table 1 Continued on next page*

*Appendix 1—table 1 Continued*

| Oligonucleotide | Sequence |
| --- | --- |
| *D2Mit165* F | TTTGGTCTTTCTAACCTTTGCA |
| *D2Mit165* R | AACAAAAACAAAACCAAAAAAACC |
| *D2Mit194* F | TGGAATTCCAAAGTCAAGGG |
| *D2Mit194* R | GGGAAGAATGGGGGAAGTTA |
| *D2Mit168* F | CTCACAGACACTGCACTATTACACA |
| *D2Mit168* R | TGTTCCTGCTATTGTTTTGGG |
| *D2Mit22* F | GCTCCCTTTCCTCTTGAACC |
| *D2Mit22* R | GGGCCCTTATTCTATCTCCC |
| *D2Mit309* F | ACAAATGCCACTCTCACATCC |
| *D2Mit309* R | TATTTCTCAGAGTCACTAGGAGTGATG |
| *D2Mit285* F | TCAATCCCTGTCTGTGGTAGG |
| *D2Mit285* R | TATGACACTTACAAGGTTTTTGGTG |
| *D2Mit48* F | GCTCTGCAGAAGATGCTGC |
| *D2Mit48* R | GCTGAGACGCAGAGTCGC |
| *D2Mit311* F | ACAGGCAGCCTTCCCTTC |
| *D2Mit311* R | TCTGTCCCGCTTCTGTTTCT |
| *D2Mit265* F | AATAATAATCAAGGTTGTCATTGAACC |
| *D2Mit265* R | TAGTCAAAATTCTTTTGTGTGTTGC |
| *D2Mit148* F | GTTCTCTGATCTACGGGCATG |
| *D2Mit148* R | TTCACTTCTACAAGTTCTACAAGTTCC |

