## [Editor Report]

In this fundamental contribution, the authors report that a widely used knockout mouse for *Mertk* carries multiple additional changes in its genome, affecting the expression of a number of genes besides *Mertk*. Notably, they show that, although the line was backcrossed to the C57 background, these changes are due to the original 129P2 genome of the embryonic stem cells in which the knockout was originally created and that through the generation of two new knockout mouse strains, in C57 embryonic stem cells, only part of the phenotype of the original *Mertk* knockout mouse can be reproduced. These important and compelling data raise awareness as to the limitations of the *Mertk*
^-/- v1^ model and limit direct inference of *Mertk*
^-/-v1^-observed phenotypes to *Mertk* deficiency alone.

---

## [Decision Letter]

**Decision letter after peer review:**

Thank you for submitting your article "Tissue-specific modifier alleles determine *Mertk* loss-of-function traits" for consideration by *eLife*. Your article has been reviewed by 2 peer reviewers, and the evaluation has been overseen by a Reviewing Editor and Satyajit Rath as the Senior Editor. The following individuals involved in the review of your submission have agreed to reveal their identity: Hind Medyouf (Reviewer #1); Marten A Hoeksema (Reviewer #2).

Essential revisions:

Textual and figure modifications are only required for clarity as recommended by the reviewers below.

*Reviewer #1 (Recommendations for the authors):*

As mentioned in the public review, TYRO3 protein levels appear to be elevated in Mertk-/- v2 or Mertk-/- v3 compared to Mertk-/- v1 (readily notable in the WB presented Figure 5a) although no significant changes are reported at mRNA. This point should be re-evaluated by the authors and discussed in their conclusions as a compensatory increase in TYRO3 may, at least in part, contribute to masking some of the Mertk-/- v2 or Mertk-/- v3 phenotypes, especially given previous work, demonstrating that even hypomorphic expression of TYRO3 in Tyro3 B6/129 can suppress the phenotypes of Mertk-/- v1 loss-of-function. WB quantification of Figure 5a may help solve this apparent discrepancy between the data presented in the figure and the main text in which it is stated that " TYRO3 levels in these newly generated mice are comparable to BL6".

Can the authors provide additional data as to the cellular mediators of the protective phenotype observed in Mertk-/-v1? What is the status of these functionally relevant cell types in the newly generated Mertk deficiency models, Mertk-/- v2, Mertk-/- v3 at steady state and upon tumor challenge?

---

## [Author Response]

Reviewer #1 (Recommendations for the authors):As mentioned in the public review, TYRO3 protein levels appear to be elevated in Mertk-/- v2 or Mertk-/- v3 compared to Mertk-/- v1 (readily notable in the WB presented Figure 5a) although no significant changes are reported at mRNA. This point should be re-evaluated by the authors and discussed in their conclusions as a compensatory increase in TYRO3 may, at least in part, contribute to masking some of the Mertk-/- v2 or Mertk-/- v3 phenotypes, especially given previous work, demonstrating that even hypomorphic expression of TYRO3 in Tyro3 B6/129 can suppress the phenotypes of Mertk-/- v1 loss-of-function. WB quantification of Figure 5a may help solve this apparent discrepancy between the data presented in the figure and the main text in which it is stated that " TYRO3 levels in these newly generated mice are comparable to BL6".

We sincerely apologize for the confusion. Quantitation of 5 independent immunoblots for TYRO3 from Mertk ^-/-V2^ and Mertk ^-/-V3^ RPEs failed to show statistically significant differences. These results are consistent with the mRNA expression data, which also failed to reveal statistically significant differences in Tyro3 expression between WT, Mertk ^-/-V2^ and Mertk ^-/-V3^ RPEs. Only Mertk ^-/-V1^ RPEs showed significantly less Tyro3 mRNA and protein compared to WT.

Can the authors provide additional data as to the cellular mediators of the protective phenotype observed in Mertk-/-v1? What is the status of these functionally relevant cell types in the newly generated Mertk deficiency models, Mertk-/- v2, Mertk-/- v3 at steady state and upon tumor challenge?

This is an excellent question. Understanding the cellular and molecular basis of anti-tumor protection in Mertk ^-/-V1^ would be highly significant. We have been indeed pursuing this remarkably important but extremely complex question. So far, we have been able to determine that the anti-tumor resistance is entirely dependent upon T cells. Depletion of CD8^+^ or CD4^+^ T cells completely reverts this anti-tumor phenotype in Mertk ^-/-V1^. Furthermore, we have performed extensive bone-marrow chimera experiments followed by single cell RNA sequencing to individually compare and contrast the hematopoietic and stromal compartments of Mertk ^-/-V1^ with WT B6 in tumor implanted mice, as well as performed linkage mapping to investigate the Mendelian inheritance of the element that determines the anti-tumor immune phenotype. We are currently in the process of preparing a separate manuscript to describe these results. We hope that the Reviewer will agree that these findings are beyond the scope of this manuscript.